



Sequential Nutrient Uptake by Phytoplankton Maintains High Primary Productivity and
Balanced Nutrient Stoichiometry
Kedong Yin[1,2] and Paul J. Harrison[3]
[1]{School of Marine Sciences, Sun Yat-sen University, Guangzhou, China}
[2]{Key Laboratory of Marine Resources and Coastal Engineering in Guangdong
Province,Guangzhou, China}
[3]{Department of Earth and Ocean Sciences, University of British Columbia, Vancouver
BC V6T 1Z4}
Correspondence to: Kedong Yin, School of Marine Science, Sun Yat-sen University (East
Campus), Guangzhou Higher Education Mega Center, Guangzhou, 510006, China.
Tel. +86 (0)20 3933 6536; Fax +86 (0)20 3933 6607. E-mail yinkd@mail.sysu.edu.cn
Running head: sequential nutrient uptake, nutritional strategy, nutrient stoichiometry



## Abstract

We hypothesize that phytoplankton have the sequential nutrient uptake strategy in
order to maintain nutrient stoichiometry and high primary productivity in the water column.
Nutrient limited phytoplankton are capable of taking up the limiting nutrient first and they
take up non-limiting nutrients when the limiting nutrient debt has been overcome. We used
high resolution continuous vertical profiles of nutrients, nutrient ratios and on-board ship
incubation experiments to test this hypothesis in the Strait of Georgia. At the surface in
summer, ambient $NO_3^-$ was depleted with excess $PO_4^{3-}$ and $SiO_4^{4-}$ remaining, and as a result,
both N:P and N:Si ratios were low.  The two ratios increased to about 10:1 and 0.45:1,
respectively, at 20 m. Time series of vertical profiles showed that the leftover $PO_4^{3-}$ continued
to be removed, resulting in additional phosphorus storage by phytoplankton. There were
various shapes of vertical profiles of N:P and at the nutricline it changed quickly in response
to mixing events. A field incubation of seawater also demonstrated the sequential uptake of
$NO3^-$ (the most limiting nutrient) and then $PO_4^{3-}$ and $SiO_4^{4-}$ (the non-limiting nutrients). This
sequential uptake strategy allows phytoplankton to acquire additional cellular phosphorus and
silicon when they are available and wait for nitrogen to become available through frequent
mixing of $NO_3^-$ (or pulsed regenerated $NH_4$). Thus, phytoplankton show variability of
nutrient stoichiometry and are capable of maintaining high productivity by taking advantage
of vigorous mixing regimes. To our knowledge, this is the first study to show the dynamics of
continuous vertical profiles of N:P and N:Si ratios and to examine the responses of
phytoplankton to nutrients supplied naturally by mixing events. The continuous nutrient
profiles provided insight into the in situ dynamics of nutrient stoichiometry in the water
column and the transient status of nutrient stoichiometry of phytoplankton in the field.




## 1. Introduction

The stoichiometry of the C:N:P Redfield ratio is an average across a wide range of species and environmental conditions and remains a central tenet in oceanography as it couples ecosystem processes with ocean biogeochemistry, which is driven by physical processes in oceans (Redfield, 1958). Four mechanisms have been proposed to explain the variability in C:N:P ratios in marine plankton, as summarized by Weber and Deutsch (2010). The first mechanism emphasizes the relationship between cellular elemental stoichiometry of phytoplankton and ambient nutrient ratios, i.e., the stoichiometry of the water column. Laboratory cultures of phytoplankton that are in the steady state usually display variable cellular N:P ratios with the nutrient N:P supply ratios. Based on the average Redfield ratio, this mechanism has been used to infer the most limiting nutrient for phytoplankton and to debate which nutrient, nitrogen or phosphorus, should be managed to control eutrophication effects (Conley et al., 2009). The second mechanism suggests that the elemental stoichiometry is taxonomy specific. Diatoms were reported to drawdown nutrients with a low nutrient C:P and N:P ratios (Geider and La Roche, 2002; Elser et al., 2003; Price, 2005), while marine cyanobacteria have higher C:P and N:P ratios (Karl et al., 2001; Bertilsson et al., 2003). Based on the resource allocation theory, the third mechanism proposed the "growth rate hypothesis", which states that the elemental stoichiometry within a cell is controlled by the biochemical allocation of resources to different growth strategies (Falkowski, 2000; Elser et al., 2003; Klausmeier et al., 2004). Fast-growing cells may have a lower N:P ratio due to a larger allocation to P-rich assembly machinery of ribosomes (Loladze and Elser, 2011), whereas competitive equilibrium favors a greater allocation to P-poor resource acquisition machinery and therefore, higher N:P ratios. The fourth mechanism is related to the interference from dead plankton or organic detritus on the measurement of elemental composition of organic matter, which cannot be supported due to lack of such



measurements in oceans and coastal waters.

In culture experiments, sequential uptake of nutrients has been demonstrated for

diatoms under N and Si limitation (Conway et al., 1976; Conway and Harrison, 1977;
Harrison et al., 1989). Surge uptake of the limiting nutrient occurs when it is added to the
culture, while the uptake of the non-limiting nutrient is slowed or stopped until the diatom
has overcome its nutrient debt. Hence, the sequence of which nutrient is taken up first is
directly related to the nutrient status of the phytoplankton. It is difficult to assess the
nutritional status of phytoplankton in the field, but the application of laboratory results to the
interpretation of vertical nutrient profiles can provide information on their nutritional status.
To date, there have been no studies of sequential uptake of nutrients in the field using a series
of high resolution vertical profiles of nutrients and their application to nutritional status of the
phytoplankton.

In this study, we used high resolution continuous vertical profiles of N:P and N:Si

ratios to examine how N:P and N:Si ratios respond to the mixing in a highly dynamic coastal
water column and the uptake of nutrients. On-board ship incubation experiments were
conducted to support the observations of changes in vertical profiles of N:P and N:Si ratios.
We constructed seven conceptual profiles to illustrate how a vertical profile of N:P ratios
changes with mixing and uptake of nitrogen and phosphorus and how they could indicate the
nutritional status of the phytoplankton assemblage. The model also explains how N:P ratios
respond to mixing, particularly at the nutriclines (nitracline for $NO_3^-$, phosphacline for $PO_4^{3-}$
and silicacline for $SiO_4^{4-}$), and indicates which nutrient, $NO_3^-$ or $PO_4^{3-}$, is taken up first in the
water column. To our knowledge, this is the first study to show the dynamics of continuous
vertical profiles of N:P and N:Si ratios and to examine the nutritional status of phytoplankton
and their response to the supply of nutrients from water column mixing. We believe that our
approach can add a new dimension to examining the in situ dynamics of nutrients in the water



column and illustrate the ecological role of phytoplankton stoichiometry in phytoplankton
completion for nutrients.

**1.1. Conceptual Model of Variability in Vertical N:P ratios (Fig. 1)**
The Strait of Georgia (hereafter the Strait) is an inland sea that lies between Vancouver
Island and the mainland of British Columbia. It is an ideal area for studying the interactions
between mixing, nutrient vertical profiles and phytoplankton nutrient uptake because of its
relatively high biomass, frequent wind mixing and shallow (15 m) photic zone. The Strait is
biologically productive, but inorganic nitrogen is often undetectable in productive seasons in
the surface layer. The nutricline sitting within the euphotic zone is often associated with the
pycnocline.In the Strait, the ambient N:P ratio is ~10:1, similar to other coastal areas (Hecky
and Kilham, 1988).
We selected seven (T0 to T6) conceptual vertical profiles that we encountered in our
field studies and suggest events that likely occurred to produce these nutrient profiles (Fig. 1).
**T0:** In winter or after a strong wind event, the water column is homogeneously mixed,
and $NO_3^-$ and $PO_4^{3-}$ are uniformly distributed in the water column. **T1:** With the onset of
stratification, $NO_3^-$ and $PO_4^{3-}$ are taken up within the mixed layer. Assuming that the average
nutrient uptake ratio is N16:1P, a N:P uptake ratio that is >10:1 would decrease the ambient
N:P ratio to <10:1. **T2:** The uptake of $NO_3^-$ and $PO_4^{3-}$ proceeds at a N:P ratio >10:1 until $NO_3^-$
is just depleted. At this time the N:P ratio is near 0 and some phosphate remains in the water
column. **T3:** The remaining phosphate is completely taken up and stored as extra/surplus
intracellular phosphate. **T4:** After cross-pycnocline mixing occurs, the ambient N:P ratio in
the newly mixed water should be the same as the ratio in the deep water. As a result, the
vertical profile of the N:P ratio will form a right angle on the top part of the nutricline. **T5:**
Depending on how long the phytoplankton are nutrient limited, their response to the mixed



limiting nutrient can be different. When N deficient phytoplankton take up N only, the curve
of the N:P ratio parallels the $NO_3^-$ distribution curve and $PO_4^{3-}$ is left behind in the water
column. **T6:** On the other hand, if phytoplankton take up $PO_4^{3-}$ before $NO_3^-$ (e.g. if
phytoplankton were severely N starved, and there is a lag in $NO_3^-$ uptake), the N:P ratio
would be higher at the nutricline than below.
Similarly, this conceptual model can be applied to N, $SiO_4^{4-}$ and N:Si ratios. The
ambient (N:Si) ratio is about 0.5:1 at 20 m in the Strait, with 20 μM $NO_3^-$ and 40 μM $SiO_4^{4-}$.
As the average uptake ratio of N:Si is about 0.7-1:1 (equivalent to Si:N = 1.5-1:1)
(Brzezinski, 1985), the N:Si ratio decreases with depth. $SiO_4^{4-}$ is rarely depleted and
therefore, the N:Si ratio is mainly determined by the distribution of $NO_3^-$. The continuous
uptake of $SiO_4^{4-}$ without the uptake of $NO_3^-$ can be inferred based on the comparison between
the gradient of N:Si and the silicacline. For example, a sharper gradient of the N:Si ratio than
the silicacline would indicate the continuous uptake of $SiO_4^-$ without the uptake of $NO_3^-$ as in
T5 (Fig. 1)
**2. Materials and Methods**
**2.1. Station Locations**
The transect started from station S2, 8 km beyond the Fraser River mouth and under
the influence of the river plume and extended 108 km NW to S1 (well beyond the plume) in
the Strait of Georgia (Fig. 2). The station numbers are consistent with previous studies (Yin et
al., 1997a, b and c).
**2.2. Sampling and Data Processing**
The sampling was designed to investigate the distribution of nutrients ($NO_3^-$, $PO_4^{3-}$
and $SiO_4^-$) and N:P and N:Si ratios associated with mixing processes during August 6-14,
1991. Data at either an anchored station for 24 h, or a transect of a few stations within 10 h
was used. At each station, a vertical profile (0-25 m) of temperature, salinity, *in vivo*



fluorescence and selected nutrients (nitrate+nitrite, phosphate, silicate) were obtained. Only
vertical profiles of nutrients are presented in this study. Other data (salinity, temperature and
florescence) are published elsewhere (Yin et al., 1997a). The vertical profiling system has
been described in detail by Jones et al. (1991) and Yin et al. (1995a). Basically, a hose
connected to a water pump on deck was attached to the CTD probe or S4 (InterOcean$^{®}$)
which has the dual function of a CTD probe and a current meter.  Seawater from the pump
was connected into the sampling tubing of an AutoAnalyzer$^{®}$ on board ship for *in situ*
nutrient measurements, while the CTD probe was lowered slowly into the water at 1 m min$^{-1}$.
Each sampling produced a high resolution continuous vertical profile of physical and
biological parameters and thus the relationship between these parameters in the water column
can be easily recognized. Data from a vertical profile (a datum point every 3 s) were
smoothed over 15 s intervals.  This smoothing reduced the fluctuations caused by ship's
motion.
**2.3.    Analysis of Nutrients**

All nutrients were determined using a Technicon AutoAnalyzer II. Salinity effects on

nutrient analyses were tested on board ship and were found to be small. Therefore, no
correction was made for salinity effects. Nitrate (plus nitrite) and phosphate were determined
following the procedures of Wood et al. (1967) and Hager et al. (1968), respectively. The
analysis of silicate was based on Armstrong et al. (1967).
**2.4.  Field Incubation Experiments**

Niskin bottles (5 L) were used to take seawater samples and the samples were

transferred to acid cleaned carboys (10 L). Subsamples of seawater were transferred to
transparent polycarbonate flasks (1 L) and placed in Plexiglas tanks. The tanks were kept at
the same temperature as the surface water by pumping seawater (from the ship's intake at 3
m) through the tank. The incubation flasks were wrapped with 1 or 4 layers of neutral density





screening which corresponded to the light intensity from which the samples were taken (1 or
16 m). In the nutrient enrichment experiments, $NO_3^-$, $PO_4^{3-}$ and $SiO_4^-$ were added to the
samples, yielding final concentrations of 20-30, 2-3 and 20-30 µM, respectively.  The
incubations lasted for 24 to 96 h, and samples were taken every 3-6 h for nutrients.
**3. Results**

**3.1.  Vertical Profiles of Nutrients and Nutrient Ratios**

At S3 near the edge of the Fraser River plume, the profiles documented changes

before (T1) and after wind mixing (T7). At T1, both $NO_3^-$ and $PO_4^{3-}$ were low in the surface
layer and N:P ratios were low (<2:1) and increased to ~8:1 at 20 m (Fig. 3). At T7, higher N:P
ratios of 16-20:1 occurred due to an increase in $NO_3^-$ in the deep water. $SiO_4^{4-}$ was ~30 µM at
the surface due to input from the Fraser River, and increased to 37 µM at 20 m (Fig. 3). The
N:P ratio curve nearly formed a right angle at the top of the nutriclines when the gradient of
the nitracline was larger than that of the phosphacline. At T1, the N:Si ratio was near 0
because $NO_3^-$ was near the detection limit, but started to increase along the nitracline at the
depth of the $SiO_4^-$ minimum. At T7, N:Si increased more rapidly with the nitracline.

A strong wind event occurred on August 7 and the water column was mixed (Yin et

al., 1997a). We followed the change in the nutrient profiles and nutrient ratios from S3 near
the Fraser River plume, to P4 and P6 and the well beyond the plume to S1. At S3, N:P ratios
in the water column were >7:1 when both $NO_3^-$ and $PO_4^{3-}$ were high after wind mixing, with
N:Si ratios being <0.5:1 (Fig. 4). As the post-wind bloom of phytoplankton developed along
P4-P6 due to the newly supplied nutrients (Yin et al., 1997a), N:P ratio followed the
distribution of $NO_3^-$ at P4, and decreased to 0 as $NO_3^-$ was depleted at the surface at P6 (Fig.
4). It was clear that little $PO_4^{3-}$ was consumed while $NO_3^-$ was taken up. At the same time, the
silicacline deepened and paralleled the nitracline. At S1, N:P and N:Si ratios formed almost a
vertical line. N:P and N:Si ratios were ~8:1 and 0.5:1, respectively, in the deep water (Fig. 4).



The time series (T1, T3, T8 and T11) of Aug 8-9 captured changes over 1 or 2 days
after the wind mixing event at S1 that was well beyond the river plume (Fig. 5). At T1, N:P
and N:Si ratios were ~9:1 and 0.45:1 respectively with $NO_3^-$ and $PO_4^{3-}$ being 15 and 1.7 µM,
respectively, at the surface. At T3, N:P ratio remained constant at ~9:1, while concentrations
of $NO_3^-$ and $PO_4^{3-}$ decreased by 10 and 1 µM, respectively, indicating an uptake N:P ratio of
10:1. In comparison, N:Si ratio decreased from T1 to T3 when $SiO_4^-$ decreased by 10 µM,
producing an uptake N:Si ratio of ~1:1. At T8, N:P ratio followed the $NO_3^-$ distribution as
$NO_3^-$ decreased to ~0 µM at the surface while $PO_4^{3-}$ was still ~0.5 µM. This indicated that
$NO_3^-$ uptake was more rapid than $PO_4^{3-}$ uptake and hence $NO_3^-$ mainly determined the
ambient N:P ratios. The N:Si uptake ratio of ~1:1 continued until T8.  However, at T11, the
N:P ratio spiked higher in the top 5-10 m of the nutricline, suggesting a more rapid uptake of
$PO_4^{3-}$ relative to $NO_3^-$ in the upper portion of the phosphacline (Fig. 5).
Changes in the profiles after the wind event on Aug 7 were followed over 5 days (Aug
10 – 14) at P5 that was still within the influence of the river plume as evidenced by the higher
surface $SiO_4^{4-}$ at the surface (Fig. 6). On Aug 10-11, N:P ratios were higher at the surface
where the post-wind induced bloom occurred two days earlier, suggesting that uptake of
$PO_4^{3-}$ had caught up with uptake of $NO_3^-$. The right angle shape of the N:P ratio on Aug 12
occurred as the nutriclines became sharper due to entrainment of nutrients. By Aug 13, more
$NO_3$ was taken up at depth and the N:P ratio followed the deepening of the nitracline and
$PO_4^{3-}$ was left behind. On Aug 14, $PO_4^{3-}$ started to decrease. During Aug 10-14, a minimum
in $SiO_4^{4-}$ was present at an intermediate depth (5-10 m), coinciding with the top of the
nitracline, and the silicacline followed the nitracline below 10 m.
**3.2. Changes in Nutrient Ratios During Field Incubations**
On deck incubation experiments were used to examine changes in uptake ratios by
eliminating any effects due to mixing. Ambient N:P and N:Si ratios were lower at the surface



than at depth, indicating higher uptake of $NO_3^-$ at the surface. The indication of a higher
uptake ratio of N:P and N:Si was supported by field incubation experiments.  During nutrient
addition ($NO_3^-$, $PO_4^{3-}$ and $SiO_4^{4-}$) bioassays on a sample from 1 m at P3, all nutrients
decreased as fluorescence increased (Fig. 7). Ambient N:P and N:Si ratios decreased to
almost 0:0 after 96 h, indicating more rapid uptake of $NO_3^-$ than uptake of $PO_4^{3-}$ and $SiO_4^{4-}$.
The temporal decline in the N:P and N:Si ratios resembled the temporal progression during a
bloom as illustrated in T0-T3 of the conceptual profiles (Fig. 1) and in the water column (S3,
P4, P6) on August 8 (Fig. 4) and during the time series at S1 (Fig. 5). During the incubation,
both $PO_4^{3-}$ and $SiO_4^{4-}$ continued to be drawn down after $NO_3^-$ became undetectable (Fig. 7). In
an earlier incubation experiment at S3 near the end of the phytoplankton bloom on June 8,
$PO_4^{3-}$ was depleted at 1 m, and both $NO_3^-$ and $SiO_4^-$ continued to disappear with 2 μM $NO_3^-$
and 4 μM $SiO_4^{4-}$ being taken up. However, for the sample taken at 16 m, $PO_4^{3-}$ (~0.5 μM) and
$SiO_4^{4-}$ (~5 μM) continued to disappear after 1.25 μM $NO_3^-$ was depleted after 8 h (Fig. 8).

The water sample at S1 on June 4 was incubated for 30 h without an addition of

nutrients (Fig. 9A). The initially low $NO_3^-$, and $PO_4^{3-}$ remained near depletion levels during
the incubation, but $SiO_4^{4-}$ decreased from 9 to <1 μM (Fig. 9A), which indicated that an
additional 8 μM $SiO_4^{4-}$ was taken up in excess in relation to N and P. At the end of 30 h,
nutrients were added (Fig. 9B).  Both $NO_3^-$ and $PO_4^{3-}$ rapidly disappeared during the first 6 h,
while $SiO_4^{4-}$ decreased little (Fig. 9B), indicating a sequential uptake of $NO_3^-$ and $PO_4^{3-}$ since
8 μM $SiO_4^{4-}$ was previously taken up as shown in Fig. 9A. The N:P ratio decreased faster
after a single addition of $NO_3^-$ or $PO_4^{3-}$ alone than with additions of $NO_3^-$ and $PO_4^{3-}$ together
(Fig. 9C), suggesting an interaction between the uptake of $NO_3^-$ and $PO_4^{3-}$. The accumulative
uptake ratio of $NO_3^-$ to $PO_4^{3-}$ increased with time, especially when only a single nutrient was
present. The ratio of N:Si decreased with time, and the accumulative uptake ratio of N:Si
exceeded 3:1 in the presence of $PO_4^{3-}$ (Fig. 9C).



## 4. Discussion

The Strait is highly productive due to pulsed nutrient supplies and multiple phytoplankton blooms in the shallow photic zone interacting with wind events, and fluctuations in river discharge (Yin et al., 1997a; Yin et al., 1995c). Our results revealed sequential nutrient uptake to optimize nutrient uptake efficiency and generate high primary productivity by phytoplankton by taking advantage of pulsed nutrients in this highly dynamic relatively shallow photic zone.

### 4.1. Responses of N:P and N:Si ratios to vertical mixing and uptake of nutrients

A vertical profile of N:P and N:Si ratios represents a snapshot of the mixing and the uptake of N, P and Si by phytoplankton in the water column. The depletion zone of the most limiting nutrient in the euphotic zone ends at a depth where the uptake of nutrients just balances the upward flux of nutrients through the nutracline, as indicated in T3 in the conceptual profiles (Fig. 1). Different responses of nutrient uptake to pulsed nutrients by mixing, appear to depend on the previous stability of the water column, the depth of the euphotic zone and nutritional status of phytoplankton. Our observations spanned all seven conceptual profiles (Fig. 1) and indicated the dynamic processes influencing the sequence of nutrient uptake that is determined by the nutritional status of the phytoplankton. The change in the profiles of the N:P ratio from S3 to P6 (Fig. 4) displayed the spring bloom-like progression as illustrated in conceptual profiles of T0-T3 (Fig. 1) after the wind mixing event. Various responses of the N:P ratios were similar to the conceptual profiles T4, T5 and T6 (Fig. 1). There was a right angle pattern in the N:P ratio sitting on the top of the nutriclines at ~7 m in T7 of Fig. 3 and also at 6 m at P5 (Aug 12, Fig. 6) that was similar to the conceptual profile in T4 (Fig. 1). There were parallel lines between the nitracline and the N:P ratio curve on Aug 12, Fig. 6) that was similar to the conceptual profile in T5 (Fig. 1). At S1, there was a spike in the N:P ratio curve at T11 (Fig. 5) at the top of the nutricline due to continued uptake



of $PO_4^{3-}$ with $NO_3^-$ being depleted during the time period from T1 to T8 (Fig. 5), as illustrated
in the conceptual profile T6 (Fig. 1). The spike in the N:P ratio was continuously observed on
Aug 10 at P5 (Fig. 6).

**4.2. Sequential Nutrient Uptake for Balanced Stoichiometry and Nutritional**
**Optimization**
Phytoplankton can take advantage of the dynamic mixing regimes and optimize their
growth rates by taking up nutrients sequentially. The disappearance of nutrients during the
incubation resembled the temporal progression of a bloom as illustrated in T0-T3 of the
conceptual profiles (Fig. 1) and in the water column (S3, P4, P6; Fig. 4), or during the time
series at S1 (Fig. 5).
Nutrient deficiency results in a decrease in the cellular content of the limiting nutrient
and an increase in the cellular content of other non-limiting nutrients. Earlier studies found
that N limitation results in excess cellular content of P and Si (Conway and Harrison, 1977;
Healey, 1985; Berdalet et al., 1996). Some phytoplankton develop enhanced uptake of the
limiting nutrient such as $NH_4$ and $PO_4^{3-}$ upon its addition after a period of nutrient limitation
or starvation and there is an accompanying shut down of the non-limiting nutrient (Conway et
al., 1976; Conway and Harrison, 1977; McCarthy and Goldman, 1979). A few hours of
enhanced N uptake quickly overcomes the N debt since the enhanced uptake rate is many
times faster than the growth rate (Conway et al., 1976). For example. enhanced uptake of
phosphorus could double internal P within 5 min to 4 h depending on the degree of P
limitation and the pulsed concentration of $PO_4^{3-}$ (Healey, 1973). After the nutrient debt has
been overcome by enhanced uptake, the uptake of non-limiting nutrients returns to normal
after the cell quota of the limiting nutrient is maximal (Collos, 1986). The sequential uptake
of a limiting nutrient and then the uptake of both the non-limiting and limiting nutrient is



advantageous to allow phytoplankton to maintain maximum growth rates over several cell
generations.

### 4.3.  Significance of Sequential Uptake of Nutrients

There are two essential strategies used by phytoplankton to cope with a pulse of the

limiting nutrient (Collos, 1986). One strategy is the 'growth' response where phytoplankton
uptake of the limiting nutrient and cellular growth are coupled.  The other strategy is the
"storage" response where phytoplankton have the capability of accumulating large internal
nutrient pools, resulting in extensive uncoupling between uptake and growth, and a lag in cell
division of up to 24 h following a single addition of the limiting nutrient. The former strategy
would have the competitive advantage under frequent pulses of the limiting nutrient, whereas
the latter strategy presents an ecological advantage when the nutrient pulsing frequency is
lower than cell division rate. In the Strait, the chlorophyll maximum was frequently observed
at the nutricline (Cochlan et al., 1990; Yin et al., 1997 a, b and c). At Stn S2, there was the
chlorophyll maximum at 7 m during August 7 which contained 4 times more phytoplankton
cells than at the surface. The phytoplankton community in the chlorophyll maximum
contained diatoms such as *Chaetoceros* and *Thalassiosira* which use the 'growth' and
'storage' strategies respectively. In either case, the previous storage of non-limiting nutrients
would allow phytoplankton to utilize the limiting nutrient and thus maximize phytoplankton
growth by saving the energy expenditure associated with taking up non-limiting nutrients
under limiting irradiance. This may explain why there were various modes or patterns of the
N:P ratio at the nutricline, which indicates the different strategies of taking up nutrients
sequentially based on the nutritional status of phytoplankton. The sequential uptake strategy
allows phytoplankton to use the "storage" capacity for non-limiting nutrients and the
"growth" response for the most limiting nutrient when it becomes available by mixing
processes.



Sequential uptake of nutrients by phytoplankton can be a fundamental mechanism in
maintaining high productivity in the water column where there are frequent mixing events in
coastal waters. The sequential uptake strategy largely occurs at the nutraclines near or at the
bottom of the photic zone. There is a consistent association between the nutriclines and the
chlorophyll maximum in various aquatic environments (Cullen, 2015) and it is also common
in the Strait (Harrison et al., 1991). There is a frequent upward flux of nutrients through the
nutricline due to entrainment in the Strait (Yin et al., 1995a, b and c) and by internal waves in
the open ocean. Phytoplankton in the chlorophyll maximum are generally nutrient sufficient
and when these cells are brought up to the surface during entrainment or wind mixing (Yin et
al., 1995a), they can quickly photosynthesize (Yin et al., 1995c). When phytoplankton
exhaust the most limiting nutrient, their internal nutrient pool decreases and they sink down
to the nutriclines and take up the abundant nutrients there. Thus, the cycle of sequential
uptake of limiting and then the non-limiting nutrients may reduce nutrient deficiency in
phytoplankton.
Sequential uptake of nutrients can be an important process to maintain the
phytoplankton nutrient stoichiometry. Carbon fixation continues after a nutrient becomes
deficient (Elrifi and Turpin, 1985; Goldman and Dennett, 1985) and the storage of organic
carbon of a higher POC:N ratio is common in phytoplankton (Healey, 1973). When
phytoplankton cells with excessive organic carbon due to limitation of a nutrient, sink from
the upper euphotic zone to the nutricline where light becomes limiting, uptake of other
nutrients occurs by utilizing stored organic carbon, leading to an increase in the cellular N
and P quotas. Thus, the ratios of carbon to other nutrients approach optimum stoichiometry.
POC:N ratios at Stn S2 and S3 were observed to be between 6:1 and 7:1 in the water column,
even though both ambient $NO_3^-$ and $PO_4^{3-}$ were near detection limits (Fig. 10). This
demonstrates the lack of ambient nitrogen limitation on the cellular nutrient stoichiometry.





Even at Stn S1 where entrainment and mixing were not as strong as at Stns S2 and S3, the
POC:N ratio was only slightly higher than 7:1 (Fig. 10).
**5. Conclusion**
As summarized in the introduction, there are four mechanisms to explain the variability in
C:N:P ratios. The sequential uptake of nutrients offers another mechanism for explaining the
variability in the nutrient stoichiometry in phytoplankton in the euphotic zone. The use of
in-situ continuous vertical profiles in this showed that deficiency of a nutrient that is based
on the ambient nutrient ratio could be transient and overcome by the sequential uptake during
the nutrient mixing regimes. The sequential uptake of nutrients is an important strategy for
phytoplankton to maintain high primary productivity and near optimum cellular nutrient
stoichiometry.
**Authors contributions**
K. Yin collected data and wrote the manuscript.
PJ Harrison supported the research cruise for collection of data and designed the sampling
plan.
**Competing interests**
The authors declare that they have no conflict of interest.
**Acknowledgements**
We thank Dr. Mike St. John who coordinated the cruise. We acknowledge the Department of
Fisheries and Oceans for providing ship time, and the officers and crew of C.S.S. Vector for
their assistance. This research was funded by a Natural Sciences and Engineering Research
Council of Canada (NSERC) Strategic grant awarded to Prof. Paul J. Harrison. K. Yin
acknowledges the continuing support of NSFC 91328203 to this study.







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



**Figures captions**

Figure 1. Conceptual vertical profiles of the dynamics of N, P and N:P ratios. T0 to
T3 represent a time series nutrient uptake during bloom development and T4
to T6 indicate subsequent vertical mixing of nutrients and subsequent uptake.
The short horizontal line near the middle of the depth axis indicates the
euphotic zone depth. At T2, N disappears first and P is left which continues to
be taken up at T3. T4 represents mixing of nutrients into the bottom of the
photic zone and phytoplankton have not taken up these nutrients yet. At T5, N
is taken up first before P, while at T6, P is taken up first before N.

Figure 2. Map of the Strait of Georgia showing the study area and the sampling
stations.

Figure 3. Two vertical profiles (T1=12:15 and T7=06:15) in the time series for
August 6-7, 1991 of nutrients at S3. Left panel: $NO_3^-$, $PO_4^{3-}$ and N:P ratios.
Right panel: $SiO_4^{4-}$ and N:Si.

Figure 4. Vertical profiles at S3 near the Fraser River plume to P4 and P6 finally to
S1 that was well beyond the plume (108 km away) during August 8, 1991.
Left panel: $NO_3^-$, $PO_4^{3-}$ and N:P ratios. Right panel: $SiO_4^{4-}$ and N:Si ratios.

Figure 5. Selected vertical profiles at S1 during the time series (T1, T3, T8 and T11)
of August 8-9, 1991. Left panel: $NO_3$, $PO_4$ and N:P ratios. Right panel: $SiO_4^{4-}$
and N:Si ratios.

Figure 6. Vertical profiles in the time series at P5 during August 10-14, 1991. Left
panel: $NO_3^-$, $PO_4^{3-}$ and N:P ratios. Right panel: $SiO_4^{4-}$ and N:Si ratios.

Figure 7. Time course of duplicate in vivo fluorescence, $NO_3^-$, $PO_4^{3-}$ and $SiO_4^{4-}$, and
N:P and N:Si ratios during an in situ incubation of a water sample taken from



1 m at P3 on August 11 (11:45). $NO_3^-$, $PO_4^{3-}$ and $SiO_4^{4-}$ were added to the water sample at T=0 before the incubation.

Figure 8. Time course $NO_3^-$, $PO_4^{3-}$ and $SiO_4^-$ during the field incubation of water samples taken at Stn S3 during June 8, 1989. Top panel: sample taken at 1 m and the incubation was done under 1 layer of screening. Bottom panel: sample taken at 16 m and incubated under 4 layers of screening.

Figure 9. Time course of $NO_3^-$, $PO_4^{3-}$, and $SiO_4^{4-}$ during the field incubation of a water sample taken at Stn S1 on June 4, 1990. (A) No nutrients were added to the sample during the first 28 h; (B) nutrients were added in 8 treatments: no additions, $NO_3^-$ alone (+N), $PO_4^{3-}$ alone (P), $SiO_4^{4-}$ alone (+Si), $NO_3^-$ and $PO_4^{3-}$ together (+N+P), $NO_3^-$ and $SiO_4^{4-}$ (+N+Si), $PO_4^{3-}$ and $SiO_4^{4-}$ (+P+Si) and all three (+N+P+Si); (C) Ambient and uptake nutrient ratios calculated from the time course in (B).

Figure 10. Vertical profiles of particulate organic C:N ratios at stations Stn S2, S3 and S1 along the increasing distance from the river during August 20-23, 1990.



Fig. 1

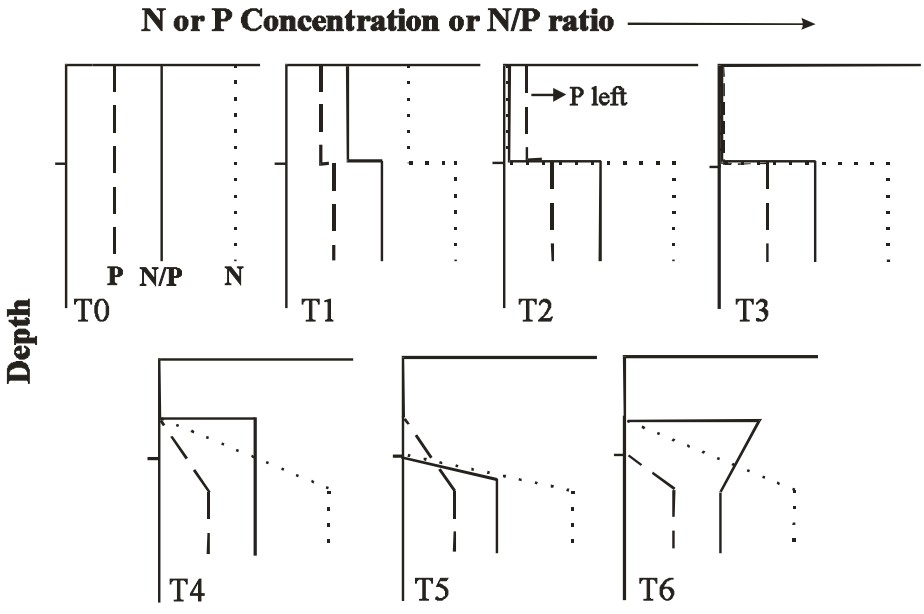





Fig. 2

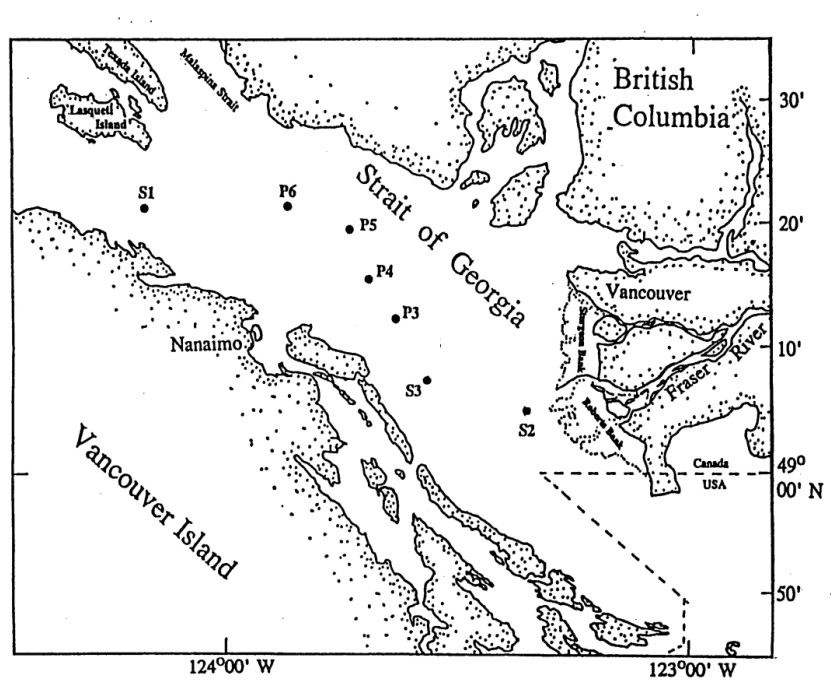



Fig. 3

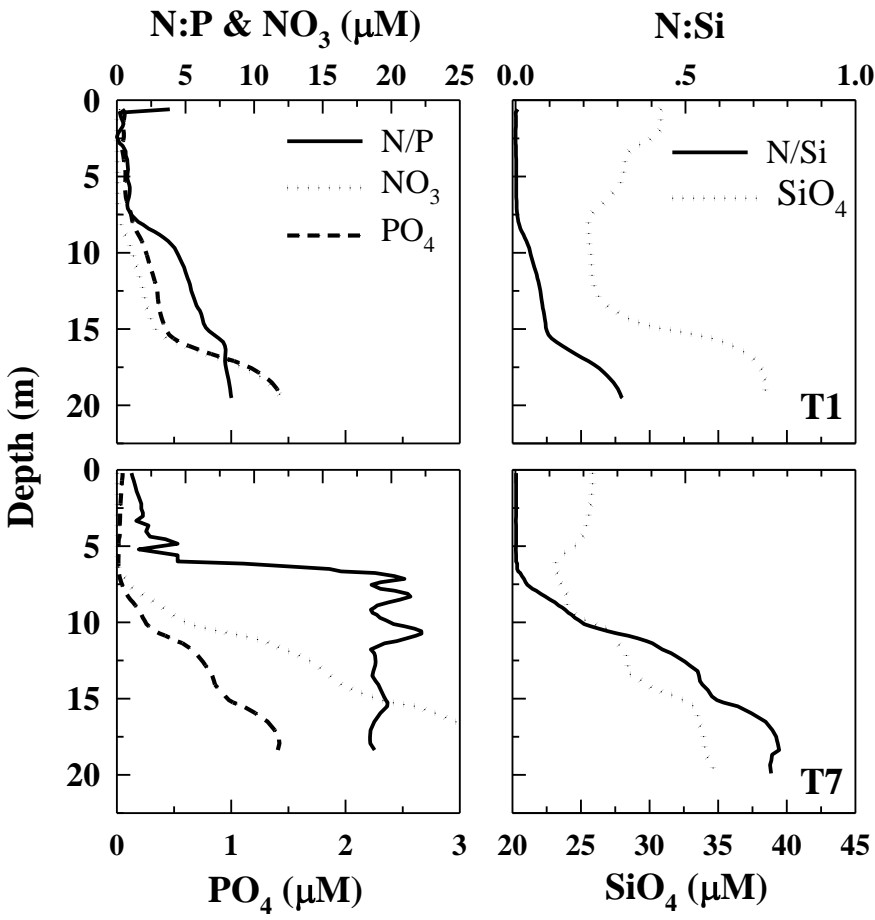





Fig. 4

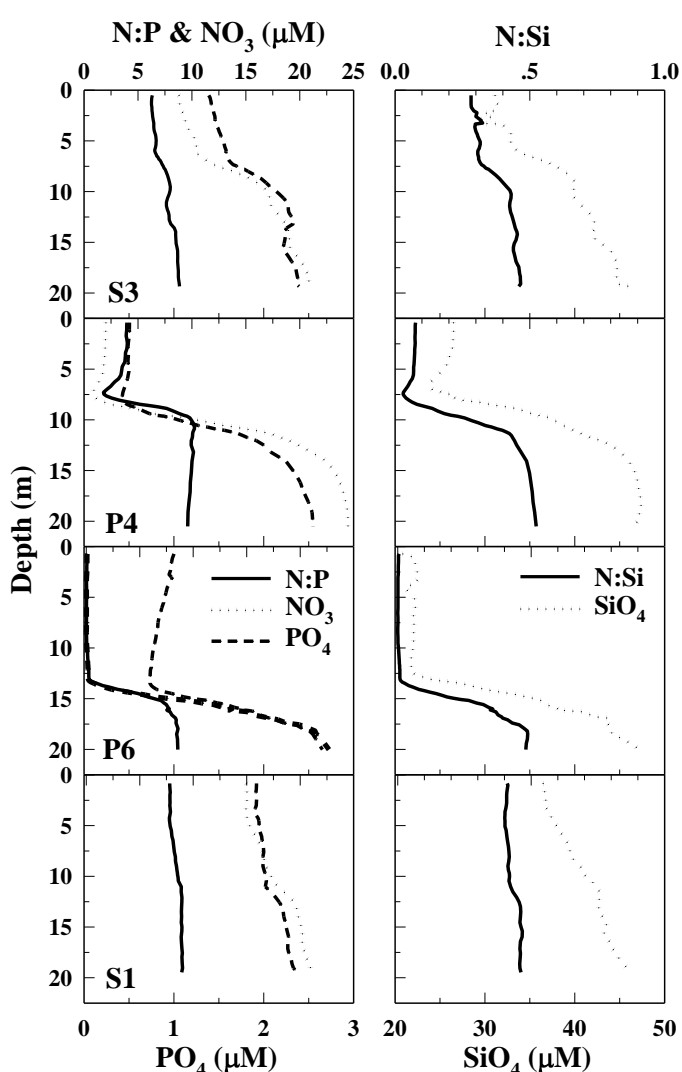



Fig. 5

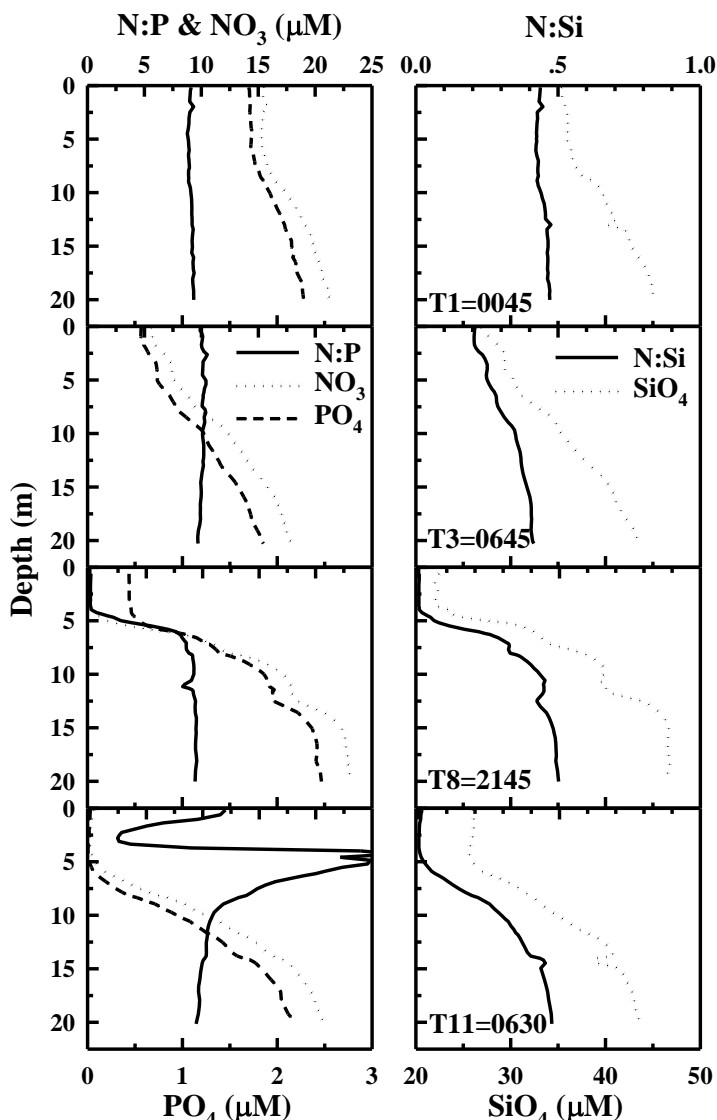




Fig. 6

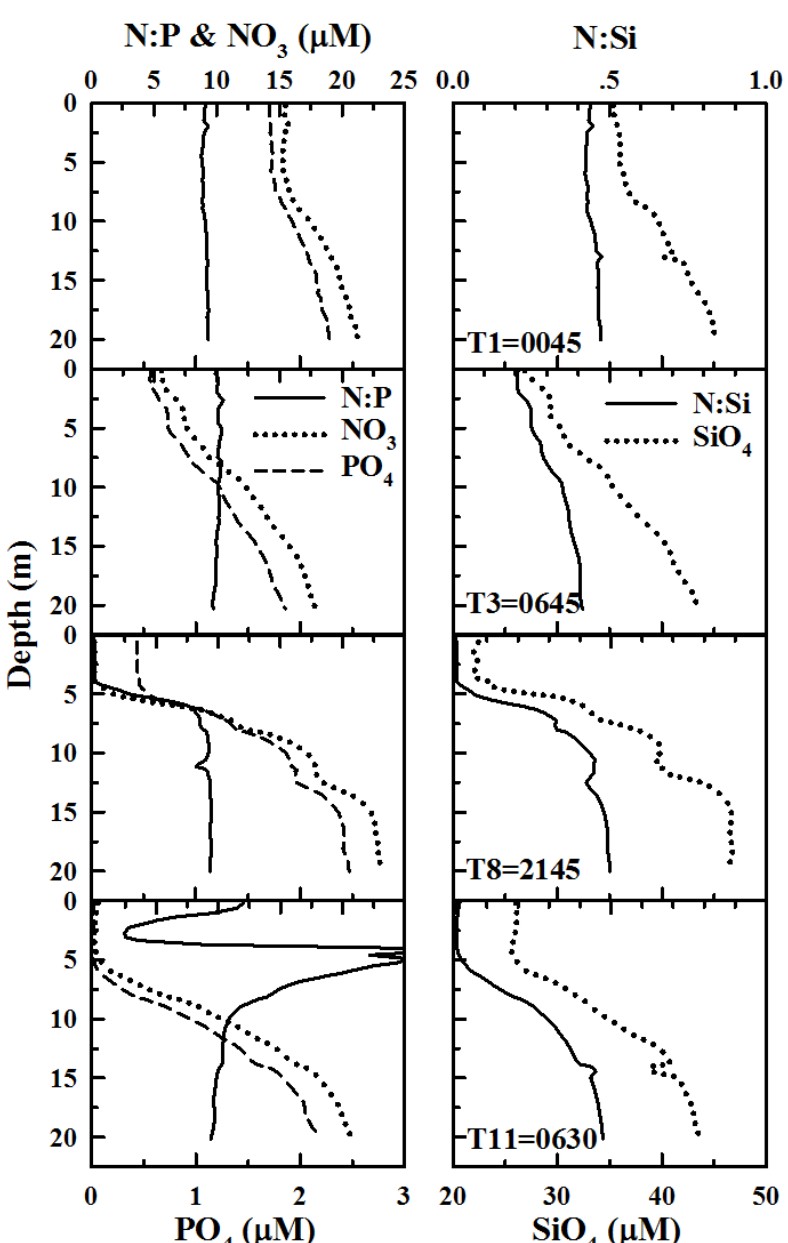



Fig. 7

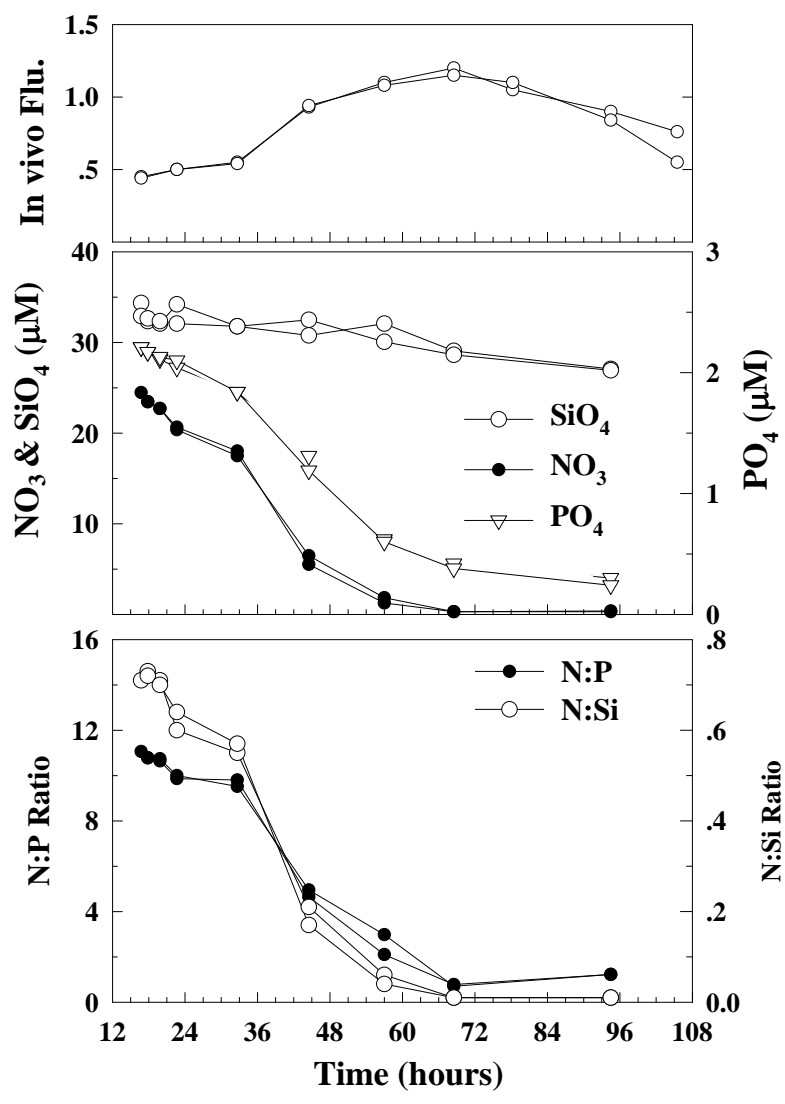



Fig. 8

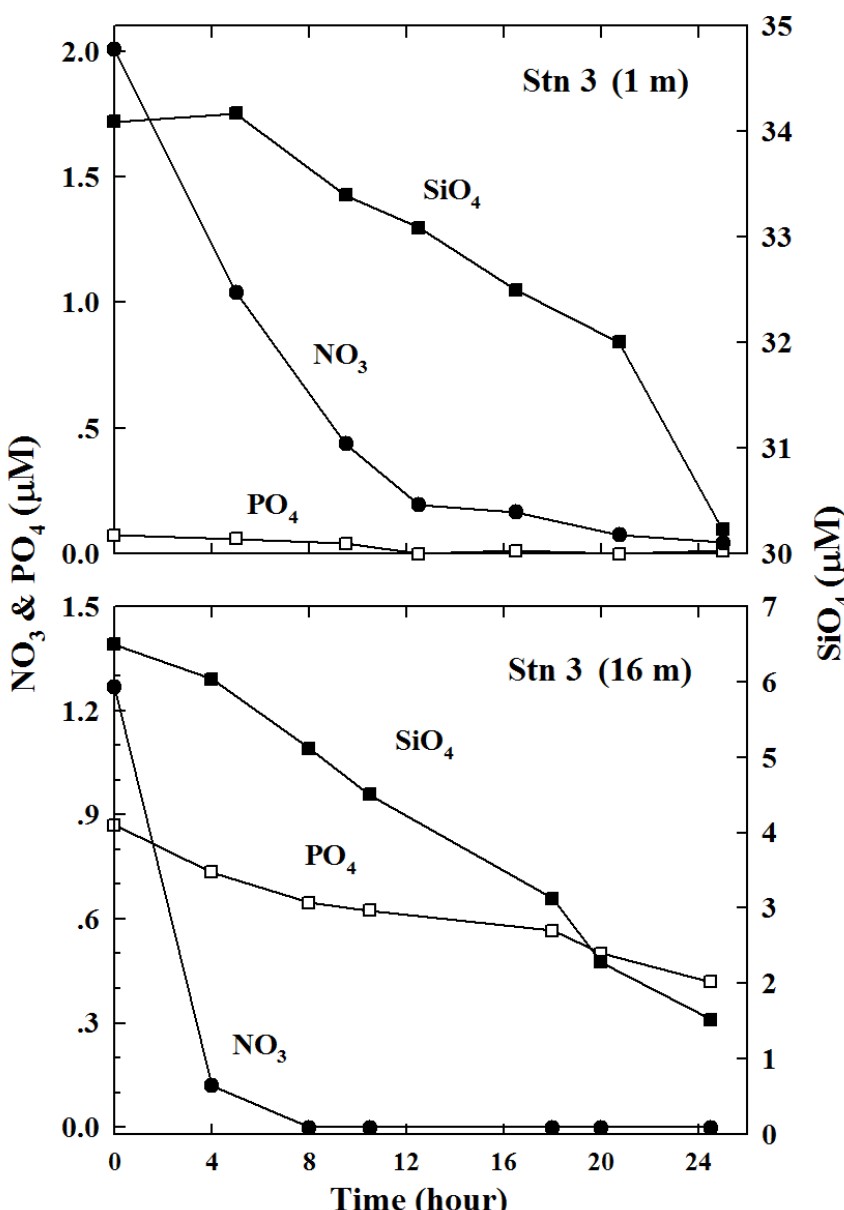



Fig. 9A

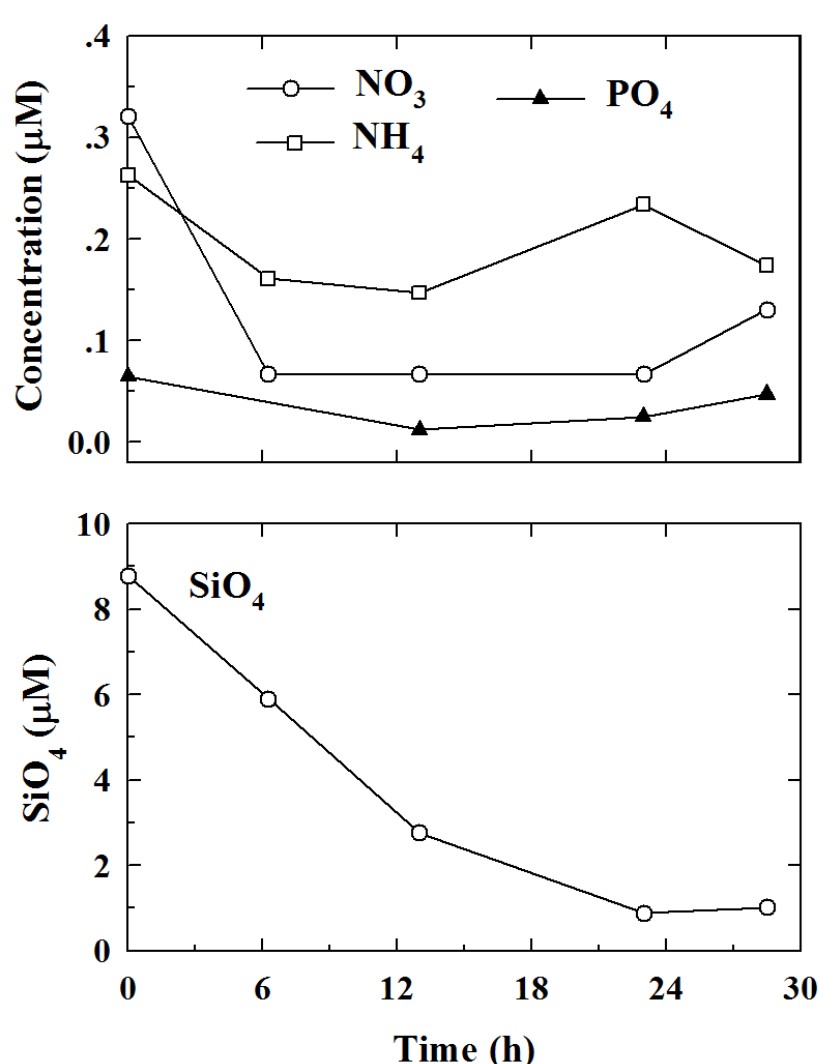



Fig. 9B

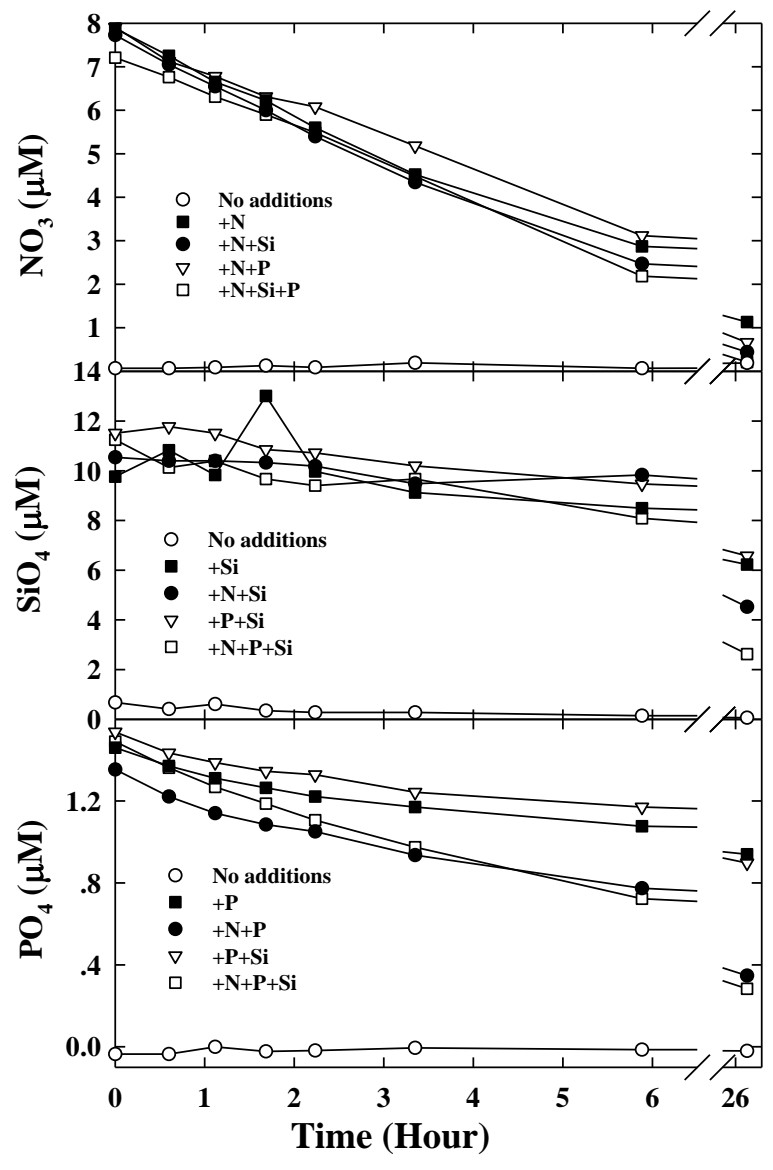





Fig. 9C

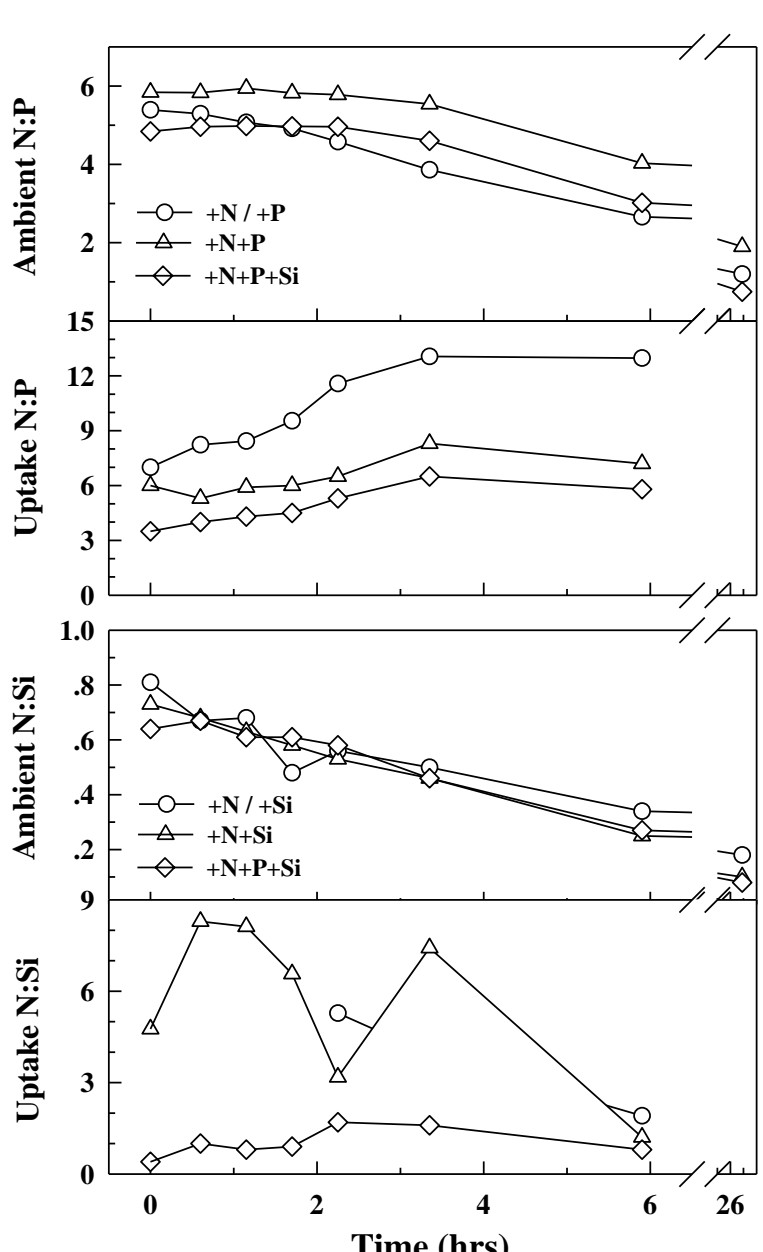



Fig. 10

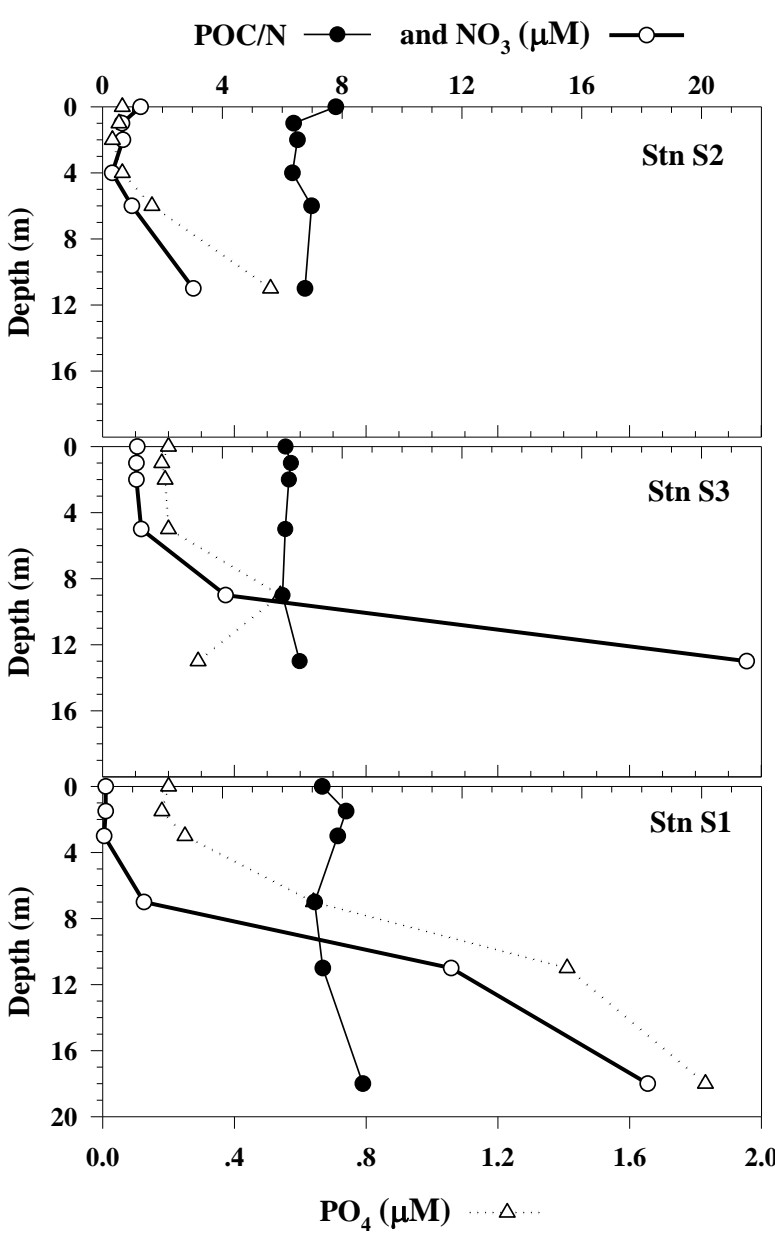