# Peer review of "Sequential Nutrient Uptake by Phytoplankton Maintains High Primary Productivity and Balanced Nutrient Stoichiometry"

_Biogeosciences, 2016_

## Referee Comment (RC1) · Anonymous Referee #1 · 22 Nov 2016

This is generally a very well written manuscript that investigates the sequential nutrient uptake strategy by phytoplankton within a coastal system to cope with maintain nutrient stoichiometry and favour growth under potentially limiting conditions. The novel use of a flow through system to sample nutrients continuously from a CTD cast allow for a uniquely high sampling resolution. The authors however rely only reporting nutrient concentrations and nutrient ratios without examining other methods for data analysis. This is particularly important for the nutrient incubation experiments that could have calculated nutrient specific growth rates. Throughout the manuscript the authors refer to high levels of primary productivity and phytoplankton growth yet fail to provide any estimates for the Strait of Georgia. The demonstration of sequential uptake by

phytoplankton to differing nutrient limitation conditions is important in understanding seasonal dynamics of productivity, community succession and nutrient concentrations. The authors mention that different uptake strategies but does suggest explicitly whether the sequential uptake favours either the growth or storage strategy.

I recommend that this manuscript be accepted; following the address of the minor revisions listed below.

Specific comments:

Page 5, Line 109: Please provide estimates of the biological productivity.

Page 6, Line 131: This paragraph gives concentrations of Nitrate and Silicate; however the previous paragraph does not give concentrations of Phosphate. If you are going to switch between a conceptual model and measured concentrations, then please be consistent and give measured concentrations for all nutrients discussed.

Page 7, Line 169: What were the detection limits of the nutrients?

Page 7, Line 170: Were the field incubations done in the same year? As the figure captions suggest they were performed in different years. There is also no mention of this when you discuss the results of these incubation experiments.

Page 9, Line 204: What was the silicate concentration at the surface? Inconsistency with the level of detail when reporting nutrient concentrations and nutrient ratios.

Page 9, Line 216: Reference to figure 6... This figure is the same as figure 5. Unable to give specific comments on the text without the correct figure to refer to. However, stylistically it would make it easier for the reader if you use the references to the time stamps in the same style as figure 5.

Page 10, Line 230: Was chlorophyll measured? Why was fluorescence not converted to chlorophyll? Increases in fluorescence do not always represent increases in biomass, but can reflect alterations to the photosynthetic apparatus; which in turn is

usually driven by the nutritional status of the phytoplankton.

Page 10, Line 251: If the diamond symbol represents the presence of phosphate, then the ratio of N:Si does not exceed 3:1 at any time point. Page 11, Line 254: 'highly productive' Once again the authors fail to give any values associated with this type of estimate.

Page 11, Line 272 – 280: This whole section reads like a re-iteration of the results without a closing statement for the reader to take away before moving onto the next section. Consider re-structuring this section.

Page 12, Line 290: 'increase in cellular content' – An increase in the cellular content of other non-limiting nutrients would only occur if luxury uptake occurs, this is not a direct result of nutrient deficiency. A direct result of nutrient deficiency is changes in intracellular nutrient stoichiometry.

Page 13, Line 324: You discuss how different phytoplankton species will either use the 'growth' 'or storage' strategies; yet here you say that phytoplankton will use 'storage' for non-limiting strategies and 'growth' for limiting nutrients. Which statement is correct? It seems like the author wants to suggest that the old idea of species specific strategies need to be revised. Suggest a bit more clarification to get this point across to the readers.

Page 14, Line 335: Can you please provide a reference for 'internal waves in the open ocean'.

Page 14, Line 335: Reference for 'Phytoplankton in the chlorophyll maximum are generally nutrient sufficient'. I don't necessarily agree with this statement; phytoplankton can exist under steady state nutrient limitation and still exist at the chlorophyll maximum within the water column.

Page 14, Line 338: How do the phytoplankton sink down? Mixing events? Changes to internal buoyancy?

Page 14, Line 350: POC/PON ratios are discussed but there is no mention to how they were measured in the methods.

Figure 1 Caption: I would suggest dropping the text that begins with 'At T2'. This reads like the discussion of the conceptual profiles that is already mentioned in the introductory text.

Figure 9A: NH4 is shown on the figure. Not mentioned in the methods or the figure caption.

Figure 9B: Symbols aren't consistent between panels making it hard to follow. i.e. Top panel, +N+P is open triangles, and then is a closed circle in the bottom panel with open triangles used for +P+Si.

Technical comments:

Page 5, Line 111: Space required between 'pynocline.' and 'In the Strait'.

Page 7, Line 153: Typo 'florescence'.

---

## Referee Comment (RC2) · Anonymous Referee #2 · 28 Nov 2016

The manuscript by Yin and Harrison measured nitrate and phosphate profiles, along with incubation experiments, to explore the ideas of nutrient drawdown in a coastal ecosystem. The title and introduction bring together ideas about the timing of nutrient uptake, the level of primary production, and how those relate to cellular nutrient stoichiometry. These are intriguing ideas and could shed light on a number of important marine processes and the linkages between them. Unfortunately, I found the presentation of methods and data to be either missing or difficult to follow. The ideas of the introduction didn't necessarily follow the data that was collected. For example, the introduction was mostly about particulate elemental ratios and diversity, but the study itself was about dissolved nutrient ratios of nitrate and phosphorus. No connection was

made between these different types of elemental ratios. Because the methods section was missing many details, it was difficult to follow what the experiments were and when they were done; therefore, it was difficult to assess the interpretation of results. I found the conceptual model presented in Figure 1 to mostly add confusion rather than clarification to the results.

There were a number of more specific issues found in the bulk of the manuscript, which have been listed below.

Suggested revisions

-Redfield is a concept for the open ocean and long-term nutrient balance with deep mixing, that specifically does not account for N-fixation or terrestrial inputs. These are not the conditions here. There is no explanation of other nitrogen forms, like ammonium and DON, which are likely important in a coastal system.

-Line 62: While the Conley et al. paper is about nutrient limitation and eutrophication control, it says nothing about Redfield, nor does it present any data. It is an opinion piece about coastal management.

-Lines 63-66: what about the work by Martiny and co-authors about global patterns of C:N:P and it's connections to diversity?

-Lines 72-75: This sentence was confusing. If the authors are stating that there are no measurements of C:N:P in heterotrophic bacteria, they should take a read through Gunderson et al. (L&O 2002) and Godwin & Cotner (ISME 2015).

-Line 138: What about the uptake of ammonium or dissolved organic nitrogen? This would certainly impact both the uptake rates and the overall drawdown of Si:N.

-The methods state that this experiment was done August 6-14, 1991, but a number of other places in the manuscript refer to additional experiments done on other dates (e.g. data shown in Figures 8 and 9). At a minimum, those additional experiments need to be described.

-For fluorescence (line 151) and nutrients (lines 165-169), more detail is needed on the standards used and detection limits.

-Line 184: Are T1 and T7 referring to time points, or conceptual models?

-Line 199: clear how? Lack of change in ambient dissolved nutrient concentrations does not necessarily imply lack of uptake. It could just as easily be fast turnover rates.

-Line 225-226: Further explanation is necessary to understand which experiments were considered "on-deck" and how that relates to the conceptual model, which is all about mixing events.

-Line 230: Fluorescence does not equal biomass.

-Lines 257-258: there is no data shown on primary production, and thus this statement is difficult to evaluate.

-Lines 269-280: The logic here is quite hard to follow, as each sentence is long and refer to multiple panels of different figures, with limited explanation and/or the use of vague terms (i.e "sitting on top" or "parallel lines").

-Line 316-317: What is the evidence for higher phytoplankton cell counts?

-Line 318-319: This statement needs to be referenced and further explained.

-Line 335-336: It's not clear how open ocean internal waves are relevant to this discussion.

-Lines 338-339: Either in this manuscript or in the literature, what evidence is there that phytoplankton are changing position in the water column in the pursuit of nutrients? The work by Bienfang and colleagues in the early '80s would indicate that physiological nutrient status does not directly correlate to sinking rates.

-Line 350: POC and PON were not discussed in the methods or results, but introduced in the discussion and figures. In addition, from looking at Figure 10, it would seem that

POC:PON ratio simply did not change, which could be due to any number of reasons, the most likely one being that C:N is a function of cell size and not limitation or luxury uptake. Besides, the introduction spells out all the reasons particulate ratios may be an unreliable measure of cellular nutrient stoichiometry.

-Lines 355-363: The conclusions don't appear to be related to the primary points in the manuscript.

-Figure 2: an inset of a larger area (zoom out) might be helpful for readers not familiar with this area. Also, the Fraser River location should be highlighted (it's a bit hard to see) and the approximate plume area/distance/direction should be indicated, as it is mentioned multiple times (e.g. lines 143, 183, 215, Figure 4, etc.) as having an influence on the sampling and results.

-Figures 5 and 6 look like copies of each other. Are the two different stations really exactly the same at all time points? Either way, what is this time series? It was not explained in the methods.

-Figure 7: The time-series results were not explained in the methods. How was this experiment performed? What is the bottom of the axis in the $NO_3^-$ (middle panel)? It looks like $NO_3^-$ goes to zero. Was the in vivo fluorescence measure calibrated to a chlorophyll standard, or was it all relative? How do the authors explain a potential lag in uptake of N and P? How would this relate to mixing events, which are presumably short-term?

-Figure 8: Is this station S3? There is no station 3 in the map in Figure 2. Why was this experiment done more than two years before the rest of the experiment? Why wasn't it explained in the methods?

-Figure 9: Most of the figure blurb needs to be in the methods. Additionally, exactly how the uptake ratios were calculated, and those results, need to be added to the manuscript. Why was this experiment done more than a year before the other experiments described herein?

-Figure 9B: This figure contains the first mention of ammonium. How (i.e. what method) was it measured?

-Figure 9C: What does the terminology of +N/+P and +N/+Si mean?

-Why was this sampling done the year prior to what was explained in the methods?

Technical revisions -Line 57: what is the "stoichiometry of the water column"? Are the authors referring to the dissolved $NO_3^-$:$PO_4$ ratio?

-Line 58-59: do the authors mean homeostatic when they say "variable"? That would make the sentence make more sense. Also, is there a reference for this relationship?

-Line 66: typo... should read "mechanism proposed is the..."

-Line 93: This should probably say that it is a "conceptual model".

-Line 101: Did the authors mean to say "competition"?

-Line 106: give a reference to Figure 2.

-Lines 113-120: It was confusing to see the conceptual models named T#, because that makes me think of a time-series. In fact, later in the paper (e.g. line 184), this same notation is used for time-series experiments.

-Line 144-145: One citation should be enough to explain station numbers.

-Why are there three figures that comprise Figure 9 given subscripts. This is a bit confusing, as lettering typically implies panels, not separate figures.

---

## Referee Comment (RC3) · Anonymous Referee #3 · 9 Dec 2016

Yin and Harrison have attempted to prove that there is preferential biological uptake of the most limiting nutrient as soon as the nutrient is added into the system. They provide high resolution nutrient data set and very interesting schematics (conceptual Fig. 1) to prove their claims. I enjoyed reading this manuscript but I still have the following suggestions that can improve the manuscript.

General comments:

1. Research in this manuscript roams around the nutrient uptake ratios. We know that the nutrient uptake and stoichiometry are phytoplankton composition dependent (see Singh et al. 2015; Mills and Arrigo 2010). Authors have not provided any cell abundance microscopic data. I understand this research was conducted long time

back but it would still improve the manuscript if authors could provide something on this aspect. They have mentioned a sentence on this in the discussion section (line 317-319) but I suggest them to add some more discussion on this.

Specific comments:

Line 38: '3' in 'nitrate' should be made subscript.

Line 103: Fig. 1 in the heading looks a bit odd

Line 111: Give space after full stop

Line 111: N:P ratio of what? of nutrients?

Line 118: Just average nutrient ratio is not 16N:1P, it is rather when averaged for all the communities together

Line 121-122: "The remaining. . . . .. . . .. . ...phosphate." Which species can take phosphate without taking any nitrate? Diazotrophs? Do they occur in the study area?

Line 175-177: "The incubation flasks. . . .. . .16m)." Mention the light intensity at 16 m, at least with compared to the surface value in terms of %. What was the euphotic depth?

Line 184: What is T7? It is not described in the conceptual model.

Line 186: "due to an increase in NO3- in the deep water", what was the source of this high nitrate? What was the station depth?

Line 187: How do the authors know that the silicate is from Fraser River? What is the silicate concentration in the river?

Line 188: "top of the nutriclines" or "top of the nutriclines at T7"

Line 192: "A strong wind", provide wind speed.

Line 220: '3' in 'nitrate' should be made subscript.

Line 235" "both. . . .. . .. . .. . .. . ...undetectable". What could be the reason for this?

In nature, who could still utilize phosphate and silicate without nitrate?

Line 249: How was the uptake ratio estimated?

Line 359: 'this' should be followed by 'study"

Line 356-363: Conclusion seems to be a bit misplaced. A lot of processes have been discussed and presented in the results but the authors have concluded only sequential uptake (which is not very convincing since there are neither any uptake measurements nor any information on community composition)

References:

Mills, Matthew M, and Kevin R Arrigo (2010) Magnitude of Oceanic Nitrogen Fixation Influenced by the Nutrient Uptake Ratio of Phytoplankton. Nature Geoscience 3(6): 412–416.

Singh, Arvind, SE Baer, Ulf Riebesell, AC Martiny, and MW Lomas (2015) C: N: P Stoichiometry at the Bermuda Atlantic Time-Series Study Station in the North Atlantic Ocean. Biogeosciences 12(21): 6389–6403.

Please also note the supplement to this comment: http://www.biogeosciences-discuss.net/bg-2016-426/bg-2016-426-RC3-supplement.pdf

---

## Author Comment (AC1) · 9 Jan 2017

Anonymous Referee #1 This is generally a very well written manuscript that investigates the sequential nutrient uptake strategy by phytoplankton within a coastal system to cope with maintain nutrient stoichiometry and favour growth under potentially limiting conditions. The novel use of a flow through system to sample nutrients continuously from a CTD cast allow for a uniquely high sampling resolution. The authors however rely only reporting nutrient concentrations and nutrient ratios without examining other methods for data analysis. This is particularly important for the nutrient incubation experiments that could have calculated nutrient specific growth rates. Throughout the manuscript the authors refer to high levels of primary productivity and phytoplankton growth yet fail to provide any estimates for the Strait of Georgia. (Addressed below with references) The demonstration of sequential uptake by phytoplankton to differing nutrient limitation conditions is important in understanding seasonal dynamics of productivity, community succession and nutrient concentrations. The authors mention that different uptake strategies but does suggest explicitly whether the sequential uptake favors either the growth or storage strategy (addressed below). I recommend that this manuscript be accepted; following the address of the minor revisions listed below. Specific comments:

–Reviewer 1: Page 5, Line 109: Please provide estimates of the biological productivity.

**Reply: Values and a reference have been added. Daily production up to 5 g C/ m2/day and annual about >300 g C/m2/yr Harrison, P.J., P.J. Clifford, K. Yin, M. St. John, M.J. Sibbald, L.J. Albright, W.P. Cochlan and P.A. Thompson. Nutrient and plankton dynamics in the Fraser River plume, Strait of Georgia, British Columbia. Mar. Ecol. Prog. Ser. 70: 291-304 (1991). Harrison, P.J., T.R. Parsons, F.J.R. Taylor and J.D. Fulton. Review of Biological oceanography of the Strait of Georgia: Pelagic Environment. Can. J. Fish. Aquat. Sci. 40: 1064 1094 (1983).**

–Reviewer 1: Page 6, Line 131: This paragraph gives concentrations of Nitrate and Silicate; however the previous paragraph does not give concentrations of Phosphate. If you are going to switch between a conceptual model and measured concentrations, then please be consistent and give measured concentrations for all nutrients discussed.

**Reply: We have deleted the word "concentration" to be consistent.**

–Reviewer 1: Page 7, Line 169: What were the detection limits of the nutrients?

**Reply: NO3 = 0.1 uM, NH4 = 0.05 uM, PO4 = 0.05 uM, SiO4 = 0.01 uM**

–Reviewer 1: Page 7, Line 170: Were the field incubations done in the same year? As the figure captions suggest they were performed in different years. There is also no

mention of this when you discuss the results of these incubation experiments.

**Reply: The samples were taken in different years, but at the same time of the year. This is noted in the methods now.**

–Reviewer 1: Page 9, Line 204: What was the silicate concentration at the surface? Inconsistency with the level of detail when reporting nutrient concentrations and nutrient ratios.

**Reply: The dashed lines for silicate on Fig. 5 were very dim, especially on an Apple Mac. We have fixed this problem.**

–Reviewer 1: Page 9, Line 216: Reference to figure 6. . . This figure is the same as figure 5. Unable to give specific comments on the text without the correct figure to refer to. However, stylistically it would make it easier for the readier if you use the references to the time stamps in the same style as figure 5.

**Reply: Yes, there was a mistake with Fig. 6. Figs. 5 and 6 should be different figures. This has been fixed now. We also fixed the problem of dim dashed lines for silicate.**

–Reviewer 1: Page 10, Line 230: Was chlorophyll measured? Why was fluorescence not converted to chlorophyll? Increases in fluorescence do not always represent increases in biomass, but can reflect alterations to the photosynthetic apparatus; which in turn is usually driven by the nutritional status of the phytoplankton.

**Reply: Chlorophyll was not measured. An increase in fluorescence usually indicates the increase in biomass in waters, which do not have strong interfering substance such as high concentrations of dissolved organic matter, particularly in the initial incubation phase under sunlight.**

–Reviewer 1: Page 10, Line 251: If the diamond symbol represents the presence of phosphate, then the ratio of N:Si does not exceed 3:1 at any time point.

**Reply: Corrected. Thank you.**

–Reviewer 1: Page 11, Line 254: 'highly productive' Once again the authors fail to give any values associated with this type of estimate.

**Reply: Revised as "The Strait of Georgia is highly productive, reaching up to 2,700 mg C m-2d-1 in August. (Yin et al. 1997a)"**

–Reviewer 1: Page 11, Line 272 – 280: This whole section reads like a re-iteration of the results without a closing statement for the reader to take away before moving onto the next section. Consider re-structuring this section.

**Reply: We have revised these sentences into a sentence to summarize the value of the conceptual model to extract information from this sequence of events.**

–Reviewer 1: Page 12, Line 290: 'increase in cellular content' – An increase in the cellular content of other non-limiting nutrients would only occur if luxury uptake occurs, this is not a direct result of nutrient deficiency. A direct result of nutrient deficiency is changes in intracellular nutrient stoichiometry.

**Reply: We have revised as "Nutrient deficiency results from a decrease in the cellular content of the limiting nutrient and continuous uptake of other non-limiting nutrients."**

–Reviewer 1: Page 13, Line 324: You discuss how different phytoplankton species will either use the 'growth' 'or storage' strategies; yet here you say that phytoplankton will use 'storage' for non-limiting strategies and 'growth' for limiting nutrients. Which statement is correct? It seems like the author wants to suggest that the old idea of species specific strategies need to be vreised. Suggest a bit more clarification to get this point across to the readers.

**Reply: We have revised this section quite a bit.**

–Reviewer 1: Page 14, Line 335: Can you please provide a reference for 'internal waves in the open ocean'.

**Reply: a reference paper has been added Pomar, L., M. Morsilli, P. Hallock, B. BaÌÅ-**

denas. 2012. Internal waves, an under-explored source of turbulence events in the sedimentary record. Earth-Science Reviews 111, 56-81.

–Reviewer 1: Page 14, Line 335: Reference for 'Phytoplankton in the chlorophyll maximum are gen- erally nutrient sufficient'. I don't necessarily agree with this statement; phytoplankton can exist under steady state nutrient limitation and still exist at the chlorophyll maximum within the water column.

**Reply: Revised as "Phytoplankton in the chlorophyll maximum are frequently exposed to nutrients and . . ."**

–Reviewer 1: Page 14, Line 338: How do the phytoplankton sink down? Mixing events? Changes to internal buoyancy?

**Reply: Changes to their internal buoyancy (exchange of heavy ions for lighter ones) and also by clumping since under nutrient deficiency cells produce extracellular carbohydrates that make them sticky and prone to clumping. – Clumping added to the text.**

–Reviewer 1: Page 14, Line 350: POC/PON ratios are discussed but there is no mention to how they were measured in the methods.

**Reply: Inserted in the methods —- POC and PON in a water sample was filtered onto a GF/F filter and analyzed with a Carlo Erba model NA 1500 NCS elemental analyzer, using the dry combustion method described by Sharp (1974).**

Sharp, JH (1974) Improved analysis of particulate organic carbon and nitrogen from seawater. Limnol Oceanogr 19:984-989

–Reviewer 1: Figure 1 Caption: I would suggest dropping the text that begins with 'At T2'. This reads like the discussion of the conceptual profiles that is already mentioned in the introductory text.

**Reply: This figure is important. It will be hard for readers to go back to the text for**

explanations. Therefore, we think that we prefer to keep this legend.

–Reviewer 1: Figure 9A: NH4 is shown on the figure. Not mentioned in the methods or the figure caption.

**Reply: NH4 is now in the methods and the figure legend.**

–Reviewer 1: Figure 9B: Symbols aren't consistent between panels making it hard to follow. i.e. Top panel, +N+P is open triangles, and then is a closed circle in the bottom panel with open triangles used for +P+Si.

**Reply: The symbols are now fixed.**

–Reviewer 1: Technical comments: Page 5, Line 111: Space required between 'pynocline.' and 'In the Strait'. Page 7, Line 153: Typo 'florescence'.

**Reply: Corrected. Thank you.**

End of reply

---

## Author Comment (AC2) · 9 Jan 2017

–Reviewer 2

The manuscript by Yin and Harrison measured nitrate and phosphate profiles, along with incubation experiments, to explore the ideas of nutrient drawdown in a coastal ecosystem. The title and introduction bring together ideas about the timing of nutrient uptake, the level of primary production, and how those relate to cellular nutrient stoichiometry. These are intriguing ideas and could shed light on a number of important marine processes and the linkages between them. Unfortunately, I found the presen-

tation of methods and data to be either missing or difficult to follow. The ideas of the introduction didn't necessarily follow the data that was collected. For example, the introduction was mostly about particulate elemental ratios and diversity, but the study itself was about dissolved nutrient ratios of nitrate and phosphorus. No connection was made between these different types of elemental ratios. Because the methods section was missing many details, it was difficult to follow what the experiments were and when they were done; therefore, it was difficult to assess the interpretation of results. I found the conceptual model presented in Figure 1 to mostly add confusion rather than clarification to the results. There were a number of more specific issues found in the bulk of the manuscript, which have been listed below.

**Reply: Thank you for your comments. We have revised the manuscript based on your suggestions and comments.**

–Reviewer 2 Suggested revisions -Redfield is a concept for the open ocean and long-term nutrient balance with deep mixing, that specifically does not account for N-fixation or terrestrial inputs. These are not the conditions here. There is no explanation of other nitrogen forms, like ammonium and DON, which are likely important in a coastal system.

**Reply: Redfield ratio is also a concept for phytoplankton nutrient composition. Ammonium concentration was usually small in the Strait of Georgia during summer and was not considered to contribute so much to dissolved inorganic nitrogen. DON is not considered in this conceptual model of sequential nutrient uptake as no evidence indicate rapid uptake of DON.**

–Reviewer 2 -Line 62: While the Conley et al. paper is about nutrient limitation and eutrophication control, it says nothing about Redfield, nor does it present any data. It is an opinion piece about coastal management.

**Reply: Redfield ratio has been used to indicate which, N or P, is the most limiting nutrient that should be controlled when managing coastal eutrophication. We have**

deleted this citation as our statement is common enough.

–Reviewer 2 -Lines 63-66: what about the work by Martiny and co-authors about global patterns of C:N:P and it's connections to diversity?

**Reply: Yes, we have referred to the paper by Martiny et al. (2013, Nature Geosciences).**

–Reviewer 2 Lines 72-75: This sentence was confusing. If the authors are stating that there are no measurements of C:N:P in heterotrophic bacteria, they should take a read through Gunderson et al. (L&O 2002) and Godwin & Cotner (ISME 2015).

**Reply: We have revised the sentence. In the measurements of elemental ratios of C:N:P of organic matter, dead plankton or organic detritus can not be separated from live organisms such as bacteria and phytoplankton. Therefore, when concentrations of these non-living organic matter vary, they contribute to our measurement of elemental ratios, but it is hard to assess their relative contributions.**

–Reviewer 2 -Line 138: What about the uptake of ammonium or dissolved organic nitrogen? This would certainly impact both the uptake rates and the overall drawdown of Si:N.

**Reply: Ammonium produced by zooplankton can be taken up and affect drawdown of N:Si, but ammonium is usually very low in the Strait of Georgia during summer and its effect was assumed to be small.**

–Reviewer 2 -The methods state that this experiment was done August 6-14, 1991, but a number of other places in the manuscript refer to additional experiments done on other dates (e.g. data shown in Figures 8 and 9). At a minimum, those additional experiments need to be described.

**Reply: The incubation experiments were conducted in different years, but in the same season. We have added the description in Methods.**

–Reviewer 2 -For fluorescence (line 151) and nutrients (lines 165-169), more detail is needed on the standards used and detection limits.

**Reply: Fluorescence has a relative unit, no standardization was made. The standards of nutrients are self-made with chemicals $NaNO_3$, $NH_4Cl$, $KH_2PO_4$, $NaSiO_4$. Detection limits are as follows. $NO_3 = 0.1$ uM, $NH_4 = 0.05$ uM, $PO_4 = 0.05$ uM, $SiO_4 = 0.01$ uM**

–Reviewer 2 -Line 184: Are T1 and T7 referring to time points, or conceptual models?

**Reply: Yes, they are referring to time points, as shown in the figure legend. However, we have changed T0, T1, . . . T6 to C0, C1, . . .. C6 in Fig. 1 to avoid the confusion.**

–Reviewer 2 -Line 199: clear how? Lack of change in ambient dissolved nutrient concentrations does not necessarily imply lack of uptake. It could just as easily be fast turnover rates.

**Reply: Yes, you are right. In this case here, we stated: "little $PO_4^{3-}$ was consumed while $NO_3^-$ was taken up", which indicates that turnover of nitrogen did not stop $NO_3$ uptake so that N:P ratio followed $NO_3$.**

–Reviewer 2 -Line 225-226: Further explanation is necessary to understand which experiments were considered "on-deck" and how that relates to the conceptual model, which is all about mixing events.

**Reply: The incubation experiments conducted on board the ship were considered to be "on-deck" experiments. These experiments show that sequential nutrient uptake happens in seawater and confirm our observations of vertical profiles of N:P and N:Si ratios which are related to the conceptual model.**

–Reviewer 2 -Line 230: Fluorescence does not equal biomass.

**Reply: Yes, you are right. Here we used it for an indication of when we could stop incubation. We found that the disappearance of the most limiting nutrient usually hap-**

[Figure]

pens one day before fluorescence reaches the maximum.

–Reviewer 2 -Lines 257-258: there is no data shown on primary production, and thus this statement is difficult to evaluate.

**Reply: Revised as "The Strait of Georgia is highly productive, reaching up to 2,700 mg C m-2d-1 in August. (Yin et al. 1997a)"**

–Reviewer 2 -Lines 269-280: The logic here is quite hard to follow, as each sentence is long and refer to multiple panels of different figures, with limited explanation and/or the use of vague terms (i.e "sitting on top" or "parallel lines").

**Reply: We have revised the section to simplify the discussion.**

–Reviewer 2 -Line 316-317: What is the evidence for higher phytoplankton cell counts? -Line 318-319: This statement needs to be referenced and further explained.

**Reply: We have made references for the sentence, and also revised this paragraph based on another reviewer.**

–Reviewer 2 -Line 335-336: It's not clear how open ocean internal waves are relevant to this discussion.

**Reply: In the open oceans, there are usually a permanent feature of the subsurface chlorophyll maximum. Phytoplankton there could use the sequential nutrient uptake strategy to maintain growth. Therefore, we would like to imply that our concept of sequential nutrient uptake is widely applicable.**

–Reviewer 2 -Lines 338-339: Either in this manuscript or in the literature, what evidence is there that phytoplankton are changing position in the water column in the pursuit of nutrients? The work by Bienfang and colleagues in the early '80s would indicate that physiological nutrient status does not directly correlate to sinking rates.

**Reply: Our evidence mainly come from the vertical movement of the chlorophyll maximum. For example, in Yin et al. (1997a), we observed that the chlorophyll maximum**

was at the surface on Aug 10 and moved down to form the subsurface chorophyll maximum couples of days later. We think that this is due to phytoplankton sinking. We have revised the sentence to ".. their internal nutrient pool decreases and they sink down to the nutriclines, possibly due to the formation of clumps".

–Reviewer 2 -Line 350: POC and PON were not discussed in the methods or results, but introduced in the discussion and figures. In addition, from looking at Figure 10, it would seem that POC:PON ratio simply did not change, which could be due to any number of reasons, the most likely one being that C:N is a function of cell size and not limitation or luxury uptake. Besides, the introduction spells out all the reasons particulate ratios may be an unreliable measure of cellular nutrient stoichiometry.

**Reply: The method for POC and PON analysis has been added. POC and PON in a water sample was filtered onto a GF/F filter and analyzed with a Carlo Erba model NA 1500 NCS elemental analyzer, using the dry combustion method described by Sharp (1974). In laboratory cultures of phytoplankton, N limitation often leads to higher C:N ratio. In this study, we mainly focus on variability of ambient nutrient ratios, and little change in POC:PON simply shows that sequential uptake of nutrients can maintain phytoplankton stoichiometry.**

–Reviewer 2 -Lines 355-363: The conclusions don't appear to be related to the primary points in the manuscript.

**Reply: We have revised the conclusion.**

–Reviewer 2 -Figure 2: an inset of a larger area (zoom out) might be helpful for readers not familiar with this area. Also, the Fraser River location should be highlighted (it's a bit hard to see) and the approximate plume area/distance/direction should be indicated, as it is mentioned multiple times (e.g. lines 143, 183, 215, Figure 4, etc.) as having an influence on the sampling and results.

**Reply: This manuscript is mainly conceptual and the location of the study area is not**

too important. We have added a "Note" in the figure legend to point out the Fraser River.

–Reviewer 2 -Figures 5 and 6 look like copies of each other. Are the two different stations really exactly the same at all time points? Either way, what is this time series? It was not explained in the methods.

**Reply: Yes, there was a mistake. Now we have used the correct figures.**

–Reviewer 2 -Figure 7: The time-series results were not explained in the methods. How was this experiment performed? What is the bottom of the axis in the NO3-(middle panel)? It looks like NO3- goes to zero. Was the in vivo fluorescence measure calibrated to a chlorophyll standard, or was it all relative? How do the authors explain a potential lag in uptake of N and P? How would this relate to mixing events, which are presumably short-term?

**Reply: The time series results were referred to in lines 227-235. The method for the incubation experiment has been described in the Methods and also in the figure legend. The bottom axis for 3 panels is the same, incubation time. Yes, NO3 does go to zero. Fluorescence was not converted to chlorophyll as chl was not measured. Time lags in incubation experiments are usually associated with low biomass. However, in this case, we made 4 times sampling within 10 hours and there appeared to be little time lag as both NO3 and PO4 responded as a decrease within 10 hours. The relation between mixing events and the responses of phytoplankton in nutrient uptake can be coupled with or without time lags depending on phytoplankton nutritional status.**

–Reviewer 2 -Figure 8: Is this station S3? There is no station 3 in the map in Figure 2. Why was this experiment done more than two years before the rest of the experiment? Why wasn't it explained in the methods?

**Reply: Yes, it is S3. We conducted quite a few experiments during 1989-1992 and used this experiment to demonstrate continuous uptake of NO3 with little P at 1 m**

sample and continuous uptake of PO4 and SiO4 after NO3 depletion. We gave explanations in the figure legend.

–Reviewer 2 -Figure 9: Most of the figure blurb needs to be in the methods. Additionally, exactly how the uptake ratios were calculated, and those results, need to be added to the manuscript. Why was this experiment done more than a year before the other experiments described herein?

**Reply: We have added the figure blurb in the figure legend and described how N:P ratio was calculated, explained why the experiments were conducted in different years. The uptake ratio was directly calculated from the decreasing concentrations over time during the incubation of seawater samples, e.g., using (day 2- day 1 nitrate concentration) /(day 2-day1 phosphate concentraiton) to get N:P ratio on day 1.**

–Reviewer 2 -Figure 9B: This figure contains the first mention of ammonium. How (i.e. what method) was it measured?

**Reply: Yes, we have added the method for ammonium into the Method.**

–Reviewer 2 -Figure 9C: What does the terminology of +N/+P and +N/+Si mean?
-Why was this sampling done the year prior to what was explained in the methods?

**Reply: We have fixed these in the figure legend. The sign "+" means "added" and "+N/+P " means, the single added N over single added P.**

–Reviewer 2 Technical revisions -Line 57: what is the "stoichiometry of the water column"? Are the authors referring to the dissolved NO3-:PO4 ratio?

**Reply: Revised as stoichiometry of nutrients**

–Reviewer 2 -Line 58-59: do the authors mean homeostatic when they say "variable"? That would make the sentence make more sense. Also, is there a reference for this relationship?

**Reply: Eventually, N:P ratio is homeostatic and hence, we have added this word in**

the abstract, but here we meant that cellular N:P ratios vary with the nutrient supply N:P ratio. We have added a reference (Geider and La Roche 2002).

–Reviewer 2 -Line 66: typo. . . should read "mechanism proposed is the. . ." -Line 93: This should probably say that it is a "conceptual model". -Line 101: Did the authors mean to say "competition"?
 -Line 106: give a reference to Figure 2.

**Reply: Line 66: Revised: the proposed mechanism Line 93: Yes, added "conceptual" Line 101: replaced completion with competition Line 106: We have added a reference by LeBlond (1983).**

–Reviewer 2 -Lines 113-120: It was confusing to see the conceptual models named T#, because that makes me think of a time-series. In fact, later in the paper (e.g. line 184), this same notation is used for time-series experiments.

**Reply: We have changed T# in Fig. 1 to C**

–Reviewer 2 -Line 144-145: One citation should be enough to explain station numbers.

**Reply: We have reduced the number to 1.**

–Reviewer 2 -Why are there three figures that comprise Figure 9 given subscripts. This is a bit confusing, as lettering typically implies panels, not separate figures.

**Reply: We have revised the figure legend for Fig. 9, as Fig. 9-1, 9-2 and 9-3. Interactive comment on Biogeosciences Discuss., doi:10.5194/bg-2016-426, 2016.**

End of reply

---

## Author Comment (AC3) · 9 Jan 2017

Anonymous Referee #3 Received and published: 9 December 2016

Reviewer #3

Yin and Harrison have attempted to prove that there is preferential biological uptake of the most limiting nutrient as soon as the nutrient is added into the system. They provide high resolution nutrient data set and very interesting schematics (conceptual Fig. 1) to prove their claims. I enjoyed reading this manuscript but I still have the following suggestions that can improve the manuscript. General comments: 1. Research in this manuscript roams around the nutrient uptake ratios. We know that the nutrient uptake

and stoichiometry are phytoplankton composition dependent (see Singh et al. 2015; Mills and Arrigo 2010). Authors have not provided any cell abundance microscopic data. I understand this research was conducted long time back but it would still improve the manuscript if authors could provide something on this aspect. They have mentioned a sentence on this in the discussion section (line 317-319) but I suggest them to add some more discussion on this.

**Reply: Thank you. We have added more discussion on phytoplankton assemblage there.**

-Reviewer 3

Specific comments: Line 38: '3' in 'nitrate' should be made subscript. Line 103: Fig. 1 in the heading looks a bit odd Line 111: Give space after full stop Line 111: N:P ratio of what? of nutrients?

**Reply: Line 38ïijŇ NO3 is corrected to NO3- Line 103, removed Fig. 1 Line 111, added space Line 111, corrected as N:P ratio of nutrients**

-Reviewer 3 Line 118: Just average nutrient ratio is not 16N:1P, it is rather when averaged for all the communities together

**Reply: You are right.**

**ReplyïijŽ The idea in this manuscript is to demonstrate that uptake of non-limiting nutrients can be decoupled from the most limiting nutrient. Here it is phytoplankton assemblages that can continue to take up phosphate after nitrate in the ambient water has disappeared.**

-Reviewer 3 Line 175-177: "The incubation flasks. . .. . .16m)." Mention the light

BGD
intensity at 16 m, at least with compared to the surface value in terms of %. What was the euphotic depth?

**Reply: 4 layers neutral screening is about 12.5% light reduction. The euphotic zone could reach down to 20 m.**

-Reviewer 3 Line 184: What is T7? It is not described in the conceptual model.

**Reply: T7 here refers to the field vertical profile, not to the conceptual model. We have changed T0, T1, ... T6 to C0, C1, ... C6 in the conceptual model in Fig. 1 to avoid the confusion.**

-Reviewer 3 Line 186: "due to an increase in NO3- in the deep water", what was the source of this high nitrate? What was the station depth?

**Reply: In the Strait of Georgia, deep water has high concentrations of nutrients and is the source of high nitrate. The station depth is over 300 m**

-Reviewer 3 Line 187: How do the authors know that the silicate is from Fraser River? What is the silicate concentration in the river? #Reply: The dotted line for SiO4 in the manuscript was very dim on my Apple computer, and you may not see it clearly. SiO4 was minimal at 10 m with higher SiO4 at the surface and at the 20 m. This higher SiO4 is from the Fraser River as the River contains higher SiO4 than the seawater in the Strait of Georgia deep water.

-Reviewer 3 Line 188: "top of the nutriclines" or "top of the nutriclines at T7" Line 192: "A strong wind", provide wind speed.Í Line 220: '3' in 'nitrate' should be made subscript.

**Reply: All are corrected.**

-Reviewer 3 Line235"" both.....undetectable". What could be the reason for this? In nature, who could still utilize phosphate and silicate without nitrate?

**Reply: Phytoplankton uptake of nutrients can deplete these nutrients to undectable**
levels. You are right, phytoplankton can not utilize phosphate and silicate without nitrate, but there is a time lag between their uptake, ie, uptake of 3 nutrients can be decoupled in time. The idea of this paper is to say sequential uptake of these nutrients.

-Reviewer 3 Line 249: How was the uptake ratio estimated?ĺ

**Reply: The uptake ratio was directly calculated from the decreasing concentrations over time during the incubation of seawater samples, e.g., using (day 2- day 1 nitrate concentration) /(day 2-day1 phosphate concentration) to get N:P ratio on day 1.**

-Reviewer 3 Line 359: 'this' should be followed by 'study"

**Reply: revised**

-Reviewer 3 Line 356-363: Conclusion seems to be a bit misplaced. A lot of processes have been discussed and presented in the results but the authors have concluded only sequential uptake (which is not very convincing since there are neither any uptake measurements nor any information on community composition) #Reply: The conclusion has been revised

-Reviewer 3 References:

Mills, Matthew M, and Kevin R Arrigo (2010) Magnitude of Oceanic Nitrogen Fixation Influenced by the Nutrient Uptake Ratio of Phytoplankton. Nature Geoscience 3(6): 412–416.

Singh, Arvind, SE Baer, Ulf Riebesell, AC Martiny, and MW Lomas (2015) C: N: P Stoichiometry at the Bermuda Atlantic Time-Series Study Station in the North Atlantic Ocean. Biogeosciences 12(21): 6389–6403.

Please also note the supplement to this comment: http://www.biogeosciencesdiscuss.net/bg-2016-426/bg-2016-426-RC3- supplement.pdf

**Reply: These papers have been cited. Thank you.**

BGD

---

## Editor Decision (ED1)

Comments to article by Yiu et al BG 2016-426, E. Marañón, Associate Editor

Authorship: A new author has been introduced in the revised version of the manuscript. This is irregular and should be justified. This new author is not mentioned in the author contribution section.

The description of field work is incomplete. At the beginning of section 2, when describing the sampling area, the authors must also indicate the sampling dates for all data used in the article. If the sampling took place as part of wider programme, this must be indicated, together with references to published studies that report on other properties of the system during the same study. It would be helpful to have a table with sampling location, station names, and sampling date, instead of including part of this information in the figure legends. This would be particularly helpful to understand the sampling schedule during the time series experiments. The bottle experiments on-deck must also be better explained.

Title: The title states that sequential nutrient uptake maintains high productivity and a balanced nutrient content of phytoplankton, but the validity of these statements is not actually proven by the data. Strictly speaking, productivity (e.g. net phytoplankton biomass accumulation) has not been measured here, nor has the phytoplankton elemental composition. The title should be re-written to make it clear that this is a mechanism that is being proposed (and which certainly is consistent with some of the observations), but not a mechanism that has been observed. I suggest including in the title a phrase along the lines: 'Sequential nutrient uptake as a potential mechanism for phytoplankton to maintain...'

A similar comment can be made in relation to the sentence on lines 271-272.

Similarly, the linkage between sequential nutrient uptake and the maintenance of phytoplankton stoichiometry near Redfield values (C:N around 7), which the authors make in the last section of the Discussion (lines 366-370), is tenuous at best, given that C:N ratios in particulate matter do not reflect phytoplankton elemental composition alone. These limitations should be explicitly acknowledged.

Specific comments

L 29 Insert: 'According to this hypothesis…' (to clarify this is not yet a result)

L31 Re-write: 'These processes would result in…'

L38-39 Sentence is awkward, as it seems to refer to vertical profiles at the nutricline. Please re-write.

Line 44 Remove 'and'. The phrase 'subject to the homeostatic stoichiometry' is vague and may be confusing. In fact the work highlights the stoichiometric plasticity of phytoplankton, rather than its fine regulation.

Line 47. This second part of the sentence is incorrect: there are many studies showing the results of phytoplankton to natural nutrient pulses supplied by mixing. See for instance Glover et al J Plankton Res (2007) 29 (3): 263-274 for an open-ocean example and also the works of Jonathan Sharples and colleagues for shelf-sea examples.

L57 Recently

L58 Remove 'in these waters'

L69 re-write '…with low C:P and N:P ratios'

L81-82 Re-write to make it clear that this assessment, albeit difficult, is not imposible. There are a few examples of direct measurements of elemental ratios in situ both for bacterio- and phytoplankton. See Segura et al Plos One 2016 for a recent example and relevant references: http://dx.doi.org/10.1371/journal.pone.0154050

L105-105 Related to comment above regarding line 47, here the authors need to be careful when referring to 'nutritional status', which in this study is inferred but never measured, since there are no measurements of phytoplankton elemental ratios. Data of elemental ratios of suspended organic matter (which in any event are difficult to interpret in this context, because of the unkown influence of non-phytoplankton material) are reported in Fig. 10, but they were not obtained during the time-course experiments. The authors need to ackowledge the limitations in their approach, as they are inferring phytoplankton nutrient content (and hence nutritional status) from observations of nutrient concentration in seawater, but the latter can be affected by many other processes in addition to phytoplankton uptake alone.

L115-116 re-write: reaching a daily production of up to xxx and an anual production of up to xxx

Line 188 Indicate bottles were maintained on-deck.

L 186-187 Actual light attenuation percentages should be indicated.

L 245 Should be 0.0, not 0:0 (which suggests that both nutrients were exhausted)

Legend to Fig. 9: The labels +N/+P and +N/+Si are confusing. If, as the legend indicates, they represent the ratio of added N over added P or added N over added Si, why should they change over time? Those ratios refer to the initial nutrient amendment but once the experiment is proceeding, the only ratios one can measure are the actual nutrient ratios in the bottles (indicated by the other 2 lines). So what do the data labeled +N/+P represent?

Section 2.4 explains the experiment described by figure 7 (in this experiment, all nutrients were added together), but not the experiment described by figure 9, in which multiple treatments were used (including additions of single nutrients). This experiment should be described in the Methods section, and in particular the concentrations of each added nutrient should be indicated.

Lines 260, 264. To avoid confusion, clarify whether these ratios refer to ambient or uptake ratios. In fact, this applies to the manuscript in general, it should always be specified whether nutrients or nutrient ratios refer to ambient concentrations, uptake, or inferred phytoplankton composition.

Line 260-262. This is difficult to follow. It is stated that 'The N:P ratio decreased faster after a single addition of N or P alone than with additions of N and P together (Fig. 9-3)', but in Fig. 9-3

the treatments wich had only added N or added P are not shown. This again goes back to the problem that it is uncertain what the treatment labeled +N/+P refers to.

lines 276-277. Nutrient recycling should also be mentioned here. In particular, P is recycled much faster than N, which in turn is recycled faster than Si.

lines 366-370. As the authors know, elemental ratios of suspended organic matter are affected by the presence of non-phytoplankton material, such as detritus and heterotrophic bacteria. The contribution of these non-phytoplankton components to total POC and PON stocks can change rapidly and is quite difficult to ascertain. In this section, the authors should acknowledge this fact. The observation that C:N of suspended matter is close to Redfield would also be consistent with a high (non-Redfield) C:N in phytoplankton in combination with a substantial contribution of bacterial biomass with a low C:N ratio. The statement 'This demonstrates the lack of ambient nitrogen limitation on the cellular nutrient stoichiometry' is not warranted, as it is based on the assumption that the POC:PON ratios reflects solely the contribution of phytoplankton.

---

## Author Response (AR2)

Biogeosciences Discuss., doi:10.5194/bg-2016-426-RC1, 2016 © Author(s) 2016. CC-BY 3.0 License.

Comments to article by Yin et al BG 2016-426, E. Marañón, Associate Editor

**--Reviewer**

Authorship: A new author has been introduced in the revised version of the manuscript. This is irregular and should be justified. This new author is not mentioned in the author contribution section.

**Reply:**

In the first round, the new author contributed to the conceptual diagram intellectually and drew it, he has plotted the figures. He also searched the updated references. In the second round, he wrote the draft of the responses to reviewers and made the draft of revision.

**--Reviewer**

The description of field work is incomplete. At the beginning of section 2, when describing the sampling area, the authors must also indicate the sampling dates for all data used in the article. If the sampling took place as part of wider programme, this must be indicated, together with references to published studies that report on other properties of the system during the same study. It would be helpful to have a table with sampling location, station names, and sampling date, instead of including part of this information in the figure legends. This would be particularly helpful to understand the sampling schedule during the time series experiments. The bottle experiments on-deck must also be better explained.

**Reply:**

We have gone through all data descriptions and figures, and added a new table (Table 1.) to ensure that all sampling information is provided.

We added a couple of sentences in the end of the Method:

Vertical profiles and seawater samples for in-situ incubation which were used in this study were collected at different stations and different sampling times. Water column conditions such as salinity, temperature and fluorescence have been described in the listed publications as shown in Table 1.

**--Reviewer**

Title: The title states that sequential nutrient uptake maintains high productivity and a balanced nutrient content of phytoplankton, but the validity of these statements is not actually proven by the data. Strictly speaking, productivity (e.g. net phytoplankton biomass accumulation) has not been measured here, nor has the phytoplankton elemental composition. The title should be re-written to make it clear that this is a mechanism that is being proposed (and which certainly is consistent with some of the observations), but not a mechanism that has been observed. I suggest including in the title a phrase along the lines: 'Sequential nutrient uptake as a potential mechanism for phytoplankton to maintain...'

A similar comment can be made in relation to the sentence on lines 271-272.

**Reply:**

Revised the title as "Sequential Nutrient Uptake as a Potential Mechanism for Phytoplankton to Maintain High Primary Productivity and Balanced Nutrient Stoichiometry".

Revised as "Our results revealed sequential nutrient uptake as a potential mechanism to..."

**--Reviewer**

Similarly, the linkage between sequential nutrient uptake and the maintenance of phytoplankton stoichiometry near Redfield values (C:N around 7), which the authors make in the last section of the Discussion (lines 366-370), is tenuous at best, given that C:N ratios in particulate matter do not reflect phytoplankton elemental composition alone. These limitations should be explicitly acknowledged.

**Reply:**

**We revised as follows:**

In addition, POC:N ratio was slightly higher than 7:1 (Fig. 10) at Stn S1 where nitrogen was more frequently under detection limit than Stns S2 and S3. This might suggest the lack of ambient nitrogen limitation on the cellular nutrient stoichiometry. However, using C:N ratio in particular matter to infer the nutrient limitation has its limitation as particular C:N ratios do not necessarily reflect phytoplankton elemental composition alone, especially in estuarine influenced waters.

**Specific comments**

**--Reviewer**

L 29 Insert: 'According to this hypothesis...' (to clarify this is not yet a result)

**Reply:**

Inserted.

**--Reviewer**

L31 Re-write: 'These processes would result in...'

**Reply:**

**Revised.**

**--Reviewer**

L38-39 Sentence is awkward, as it seems to refer to vertical profiles at the nutricline. Please re- write.

**Reply:**

Re-write as "The N:P ratios at the nutricline in vertical profiles responded differently to mixing events.

Line 44 Remove 'and'. The phrase 'subject to the homeostatic stoichiometry' is vague and may be confusing. In fact the work highlights the stoichiometric plasticity of phytoplankton, rather than its fine regulation.

**Reply:**

**Revised:**

Thus, phytoplankton are able to maintain high productivity and balance nutrient stoichiometry by taking advantage of vigorous mixing regimes with the capacity of the stoichiometric plasticity.

**--Reviewer**

Line 47. This second part of the sentence is incorrect: there are many studies showing the results of phytoplankton to natural nutrient pulses supplied by mixing. See for instance Glover et al J Plankton Res (2007) 29 (3): 263-274 for an open-ocean example and also the works of Jonathan Sharples and colleagues for shelf-sea examples.

**Reply:**

**Revised:**

To our knowledge, this is the first study to show the in situ dynamics of continuous vertical profiles of N:P and N:Si ratios, which can provide insight into the in situ dynamics of nutrient stoichiometry in the water column and the inference of the transient status of phytoplankton nutrient stoichiometry in the coastal ocean.

**--Reviewer**

L57 Recently,

L58 Remove 'in these waters'

L69 re-write '...with low C:P and N:P ratios'

**Reply:**

**All corrected.**

L81-82 Re-write to make it clear that this assessment, albeit difficult, is not imposible. There are a few examples of direct measurements of elemental ratios in situ both for bacterio- and phytoplankton. See Segura et al Plos One 2016 for a recent example and relevant references: http://dx.doi.org/10.1371/journal.pone.0154050

**Reply:**

**Revised:**

The fourth mechanism is related to the interference from dead plankton or organic detritus with the measurement of elemental composition of organic matter, and such interference is difficulty to assess due to lack of the measurements of non-living organic matters in oceans and coastal waters. However, the X-ray microanalysis (XRMA) technique was recently used to produce simultaneous quotas of C, N, O, Mg, Si, P and S in single cell organisms (Segura-Noguera et al. 2016), which will not only help to understand the fourth mechanism, but also understand the variability of stoichiometry of phytoplankton in the oceans.

**--Reviewer**

L105-105 Related to comment above regarding line 47, here the authors need to be careful when referring to 'nutritional status', which in this study is inferred but never measured, since there are no measurements of phytoplankton elemental ratios. Data of elemental ratios of suspended organic matter (which in any event are difficult to interpret in this context, because of the unkown influence of non-phytoplankton material) are reported in Fig. 10, but they were not obtained during the time-course experiments. The authors need to ackowledge the limitations in their approach, as they are inferring phytoplankton nutrient content (and hence nutritional status) from observations of nutrient concentration in seawater, but the latter can be affected by many other processes in addition to phytoplankton uptake alone.

**Reply:**

Agree. Revised to:

"to examine responses of phytoplankton to the supply of nutrients from water column mixing".

**--Reviewer**

L115-116 re-write: reaching a daily production of up to xxx and an anual production of up to xxx

**Reply:**

**Corrected.**

**--Reviewer**

Line 188 Indicate bottles were maintained on-deck.

L 186-187 Actual light attenuation percentages should be indicated.

L 245 Should be 0.0, not 0:0 (which suggests that both nutrients were exhausted)

**Reply:**

Line 188, revised as, "The incubation bottles were maintained on-deck and lasted for ...".

Line 186-187, revised as: "which corresponded to the light intensity (50-6% of the surface light)"

Line 245, revised as 0.0

**--Reviewer**

Legend to Fig. 9: The labels +N/+P and +N/+Si are confusing. If, as the legend indicates, they represent the ratio of added N over added P or added N over added Si, why should they change over time? Those ratios refer to the initial nutrient amendment but once the experiment is proceeding, the only ratios one can measure are the actual nutrient ratios in the bottles (indicated by the other 2 lines). So what do the data labeled +N/+P represent?

**Reply:**

+N stands for "adding N alone" into bottle 1, then we measured concentrations of all three nutrients: NO3, PO4, and SiO4 in this bottle daily during the incubation.

+P stands for "adding P alone" into bottle 2, then we measured concentrations of all three nutrients in bottle 2 daily.

+N/+P means ambient NO3 in bottle 1/ambient PO4 in bottle 2. We measured nutrients in some time intervals, and hence, we had ambient N:P for +N/+P over time.

+N+P stands for "two nutrients in the same bottle".

**--Reviewer**

Section 2.4 explains the experiment described by figure 7 (in this experiment, all nutrients were added together), but not the experiment described by figure 9, in which multiple treatments were used (including additions of single nutrients). This experiment should be described in the Methods section, and in particular the concentrations of each added nutrient should be indicated.

**Reply:**

For experiments with additions of a single nutrient alone or multiple nutrients together, a water sample taken at Stn S1 on June 4, 1990. The sample was incubated with no nutrients being added during the first 28 h (pre-incubation); after pre-incubation, nutrients were added in 8 treatments: no additions,  $NO_3^-$  alone (+N),  $PO_4^{3-}$  alone (+P),  $SiO_4^-$  alone (+Si),  $NO_3^-$  and  $PO_4^{3-}$  together (+N+P),  $NO_3^-$  and  $SiO_4^-$  (+N+Si),  $PO_4^{3-}$  and  $SiO_4^-$  (+P+Si) and all three (+N+P+Si). The final concentrations of added  $NO_3^-$ ,  $PO_4^{3-}$  and  $SiO_4^-$  was 7-8, 1.3-1.6 and 10-12 µM, respectively.

**--Reviewer**

Lines 260, 264. To avoid confusion, clarify whether these ratios refer to ambient or uptake ratios. In fact, this applies to the manuscript in general, it should always be specified whether nutrients or nutrient ratios refer to ambient concentrations, uptake, or inferred phytoplankton composition.

**Reply:**

We have added "ambient" to wherever is applicable.

Line 260-262. This is difficult to follow. It is stated that 'The N:P ratio decreased faster after a single addition of N or P alone than with additions of N and P together (Fig. 9-3)', but in Fig. 9-3 treatments which had only added N or added P are not shown. This again goes back to the problem that it is uncertain what the treatment labeled +N/+P refers to.

**Reply:**

**Again:**

+N stands for "adding N alone" into bottle 1, then we measured concentrations of all three nutrients: NO3, PO4, and SiO4 in this bottle daily during the incubation.

+P stands for "adding P alone" into bottle 2, then we measured concentrations of all three nutrients in bottle 2 daily.

+N/+P means ambient NO3 in bottle 1/ambient PO4 in bottle 2. We measured nutrients daily, and hence, we had ambient N:P for +N/+P over days.

+N+P stands for "two nutrients in the same bottle".

**We have revised as:**

The ambient N:P ratio decreased faster in the samples with a single addition of  $NO_3^-$  or  $PO_4^{3-}$  alone (+N/+P) than that with additions of  $NO_3^-$  and  $PO_4^{3-}$  together (+N+P) (Fig. 9-3).

**--Reviewer**

lines 276-277. Nutrient recycling should also be mentioned here. In particular, P is recycled much faster than N, which in turn is recycled faster than Si.

**Reply:**

Add the following sentence in the end of the paragraph.

The recycling of nutrients in different preferences such as faster P regeneration than N, which in turn is recycled faster than Si also contributes to the variability of nutrient ratios as shown above.

lines 366-370. As the authors know, elemental ratios of suspended organic matter are affected by the presence of non-phytoplankton material, such as detritus and heterotrophic bacteria. The contribution of these non-phytoplankton components to total POC and PON stocks can change rapidly and is quite difficult to ascertain. In this section, the authors should acknowledge this fact. The observation that C:N of suspended matter is close to Redfield would also be consistent with a high (non-Redfield) C:N in phytoplankton in combination with a substantial contribution of bacterial biomass with a low C:N ratio. The statement 'This demonstrates the lack of ambient nitrogen limitation on the cellular nutrient stoichiometry' is not warranted, as it is based on the assumption that the POC:PON ratios reflects solely the contribution of phytoplankton.

**Reply:**

Yes, agree. We revised the last sentence as:

In addition, POC:N ratio was slightly higher than 7:1 (Fig. 10) at Stn S1 where nitrogen was more frequently under detection limit than Stns S2 and S3. This might suggest the lack of ambient nitrogen limitation on the cellular nutrient stoichiometry. However, using C:N ratio in particular matter to infer the nutrient limitation has its limitation as particular C:N ratios do not necessarily reflect phytoplankton elemental composition alone, especially in estuarine influenced waters.

---

## Editor Decision (ED2)

Editorial comments to article by Yin et al BG 2016-426, 2nd revision

line 195. This line says that water for incubation experiments was ('was' is missing in the text) taken on a single date, but in fact there were multiple experiments, as indicated later on line 202. Please re-write accordingly.

Legend to Fig. 9-3 reads: 'ambient nutrient ratios were calculated from measured ambient nutrients during the time course of incubation in Fig. 9-2. The sign "+" means "added". +N/+P and +N/+Si indicate the ratio of the added N alone over the added P alone and over the added Si alone, respectively.

I still think this is confusing and inaccurate, as the reader is likely to think that what is being represented is a ratio between added nutrients, instead of an ambient ratio of nutrients in the bottle.

The response of the authors makes things more puzzling:

In their response, they write: "+N/+P means ambient NO3 in bottle 1/ambient PO4 in bottle 2. We measured nutrients in some time intervals, and hence, we had ambient N:P for +N/+P over time".

It would seem that the authors are dividing the N concentration in one bottle (the one that received an addition of N) by the P concentration in a different bottle (the one that received an addition of P). But surely this 'chimera' variable is misleading. Ambient nutrient ratios should be computed for each individual bottle using only nutrient concentrations measured in the same bottle. If my intepretation of this N:P ratio labelled +N/+P is correct, then my suggestion would be to omit it, as it only makes the figure more confusing. If you decide to keep it, the figure legend must be much clearer, and explain that in the case of data labeled +N/+P the N:P ratio plotted is not actually a N:P ratio that occurred in any particular bottle, but the result of dividing N concentration in the N+ bottle by the P concentration in the P+ bottle. The same goes for the N:Si ratio and the data labelled N+/Si+.

These are non-conventional experiments, which obviously involved a lot of effort and which provide interesting results.The authors need to make an additional effor to make sure that what their report is clearly understood by the readers.

The legend to Table 1 is not formally correct. It should read something like 'Sampling stations and dates of nutrient addition incubations. The last column indicates the study where water column properties for the Strait of Georgia are reported.'

H. Liu is still not included in the author contribution section. This section must indicate the contributions of all authors.

---

## Author Response (AR3)

Editorial comments to article by Yin et al BG 2016-426, 2nd revision

Reviewer:

line 195. This line says that water for incubation experiments was ('was' is missing in the text) taken on a single date, but in fact there were multiple experiments, as indicated later on line 202. Please re-write accordingly.

Reply: we have revised the entire section of "Field Incubation Experiments"

Reviewer:

Legend to Fig. 9-3 reads: 'ambient nutrient ratios were calculated from measured ambient nutrients during the time course of incubation in Fig. 9-2. The sign "+" means "added". +N/+P and +N/+Si indicate the ratio of the added N alone over the added P alone and over the added Si alone, respectively.

I still think this is confusing and inaccurate, as the reader is likely to think that what is being represented is a ratio between added nutrients, instead of an ambient ratio of nutrients in the bottle.

The response of the authors makes things more puzzling:

In their response, they write: "+N/+P means ambient NO3 in bottle 1/ambient PO4 in bottle 2. We measured nutrients in some time intervals, and hence, we had ambient N:P for +N/+P over time".

It would seem that the authors are dividing the N concentration in one bottle (the one that received an addition of N) by the P concentration in a different bottle (the one that received an addition of P). But surely this 'chimera' variable is misleading. Ambient nutrient ratios should be computed for each individual bottle using only nutrient concentrations measured in the same bottle. If my intepretation of this N:P ratio labelled +N/+P is correct, then my suggestion would be to omit it, as it only makes the figure more confusing. If you decide to keep it, the figure legend must be much clearer, and explain that in the case of data labeled +N/+P the N:P ratio plotted is not actually a N:P ratio that occurred in any particular bottle, but the result of dividing N concentration in the N+ bottle by the P concentration in the P+ bottle. The same goes for the N:Si ratio and the data labelled N+/Si+.

These are non-conventional experiments, which obviously involved a lot of effort and which provide interesting results.The authors need to make an additional effor to make sure that what their report is clearly understood by the readers.

Reply:

As pointed out, this type of experiments is different traditional ones, and provided some information on synergistic effects of single and multiple nutrients on uptake of those nutrients. We should keep them. To avoid the confusion, we have revised the following

1. In the Materials and Methods, we have revised the nutrient addition experiments, more clearly, as follows.

after the pre-incubation during which all nutrients were depleted, nutrients were added in 8 treatments: 1) control: no additions, 2) +N: adding $NO_3^-$ alone , 3) +P:

adding $PO_4^{3-}$ alone, 4) +Si: adding $SiO_4^-$ alone, 5) +N+P: adding $NO_3^-$ and $PO_4^{3-}$ together, 6) +N+Si: adding $NO_3^-$ and $SiO_4^-$ , 7) +P+Si: adding $PO_4^{3-}$ and $SiO_4^-$ and 8) +N+P+Si: adding all three nutrients.

2. we have added new axis labels on the right side in Fig. 9-3 where we specifically spelled ambient +N/+P so that people should realize that this ratio is from different bottles +N and +P, not in the same bottle.

3. we also added a description to remind readers in the legend of Fig. 9-3, as follows. Ambient +N/+P indicates the ratio of N in +N (the added N alone) over P in +P (the added P alone). Please note that this N:P ratio is not in the same bottle. Similarly, ambient +N/+Si indicates N in +N over Si in +Si (the added Si alone).

Reviewer:

The legend to Table 1 is not formally correct. It should read something like 'Sampling stations and dates of nutrient addition incubations. The last column indicates the study where water column properties for the Strait of Georgia are reported.'

Reply:

Revised as shown in the following.

| Sampling stations | Dates of field incubation experiments | The studies that described water properties for the Strait of Georgia |
| --- | --- | --- |
|  |  |  |

Reviewer:

H. Liu is still not included in the author contribution section. This section must indicate the contributions of all authors.

Reply:

Now H. Liu has been listed and his contribution has been given.